# Nanodroplet processing platform for deep and quantitative proteome profiling of 10–100 mammalian cells

Ying Zhu[1], Paul D. Piehowski [2], Rui Zhao[1], Jing Chen [3], Yufeng Shen[2], Ronald J. Moore[2], Anil K. Shukla[2], Vladislav A. Petyuk [2], Martha Campbell-Thompson[3], Clayton E. Mathews[3], Richard D. Smith [2], Wei-Jun Qian [2] & Ryan T. Kelly [1]

Nanoscale or single-cell technologies are critical for biomedical applications. However, current mass spectrometry (MS)-based proteomic approaches require samples comprising a minimum of thousands of cells to provide in-depth profiling. Here, we report the development of a nanoPOTS (nanodroplet processing in one pot for trace samples) platform for small cell population proteomics analysis. NanoPOTS enhances the efficiency and recovery of sample processing by downscaling processing volumes to <200 nL to minimize surface losses. When combined with ultrasensitive liquid chromatography-MS, nanoPOTS allows identification of ~1500 to ~3000 proteins from ~10 to ~140 cells, respectively. By incorporating the Match Between Runs algorithm of MaxQuant, >3000 proteins are consistently identified from as few as 10 cells. Furthermore, we demonstrate quantification of ~2400 proteins from single human pancreatic islet thin sections from type 1 diabetic and control donors, illustrating the application of nanoPOTS for spatially resolved proteome measurements from clinical tissues.

---

[1] The Environmental Molecular Sciences Laboratory, Pacific Northwest National Laboratory, Richland, WA 99354 USA. [2] Biological Sciences Division, Pacific Northwest National Laboratory, Richland, WA 99354 USA. [3] Department of Pathology, Immunology and Laboratory Medicine, University of Florida, Gainesville, FL 32611 USA. Correspondence and requests for materials should be addressed to R.T.K. (email: ryan.kelly@pnnl.gov)

One of the most impactful technological advances in biological research in recent years has been the development of broad omics-based molecular profiling capabilities and their scaling to much smaller sample amounts than were previously feasible, including single cells. Highly sensitive genome amplification and sequencing techniques have been developed for the analysis of rare cell populations, interrogation of specific cells and substructures of interest within heterogeneous clinical tissues, and profiling of fine needle aspiration biopsies[1,2]. However, genomic and transcriptomic technologies only provide indirect measurements of cellular states[3]. Broad proteome measurements provide more direct characterization of phenotypes and are crucial for understanding cellular functions and regulatory networks. Flow cytometry and mass cytometry[4] approaches enable the detection of up to tens of protein markers from single cells by utilizing antibody-bound reporter species. However, these technologies are inherently limited by the availability of high-quality antibody reagents and multiplexing capacity. The biomedical field is in critical need of highly sensitive technologies for providing broad proteome measurements for very small number of cells or even single cells to enable analyses of tissue substructures, cellular microenvironments, and other applications involving rare or small subpopulations of cells.

Current mass spectrometry (MS)-based proteomic approaches are capable of providing broad measurements of protein abundances as well as post-translational modifications within complex samples. However, relatively large amounts of protein from millions of cells are typically required to achieve deep proteome coverage. Unlike genomics and transcriptomics, proteomics does not benefit from amplification strategies. Considerable efforts have thus been devoted to enhancing the overall analytical sensitivity of MS-based proteomics[5]. For example, liquid-phase separations including liquid chromatography (LC) and capillary electrophoresis have been miniaturized to reduce the total flow rate, leading to enhanced efficiencies at the electrospray ionization (ESI) source[6,7]. Advanced ion focusing approaches and optics such as the electrodynamic ion funnel[8] minimize ion losses during transfer from the atmospheric pressure ESI source to the high-vacuum mass analyzer, and are now incorporated into many advanced biological MS platforms. As a result of these and other improvements, mass detection limits as low as 10 zmol for MS and 50 zmol for tandem MS analysis of peptides have been achieved[5–7,9,10]. Conceptually, this level of analytical sensitivity is sufficient to detect many proteins at levels expressed in single mammalian cells[6,7]. However, despite this capability, application to such small samples remains largely ineffective.

The major gap between demonstrated analytical sensitivity and the present practical need for orders of magnitude more protein starting material largely derives from limitations in required sample processing, including protein extraction, proteolytic digestion, cleanup, and delivery to the analytical platform. As sample amounts decrease without a concomitant reduction in reaction volume (often limited by evaporation and the ~microliter volumes addressable by pipet), the nonspecific adsorption of proteins and peptides to the surfaces of reaction vessels, along with inefficient digestion kinetics, become increasingly problematic. Efforts to improve sample preparation procedures include the use of low-binding sample tubes and "one-pot" digestion protocols to limit total surface exposure[9,11–16]. In addition, trifluoroethanol-based protein extraction and denaturation[11], filter-aided sample preparation[12], MS-friendly surfactants[14,15], high-temperature trypsin digestion[13], adaptive focused acoustic-assisted protein extraction[9], and immobilized digestion protocols[12] have achieved some advances in the processing of small samples. Using these methods, a proteome coverage of ~600 was reported when 100 cells were analyzed, and thousands of proteins

were identified with samples comprising thousands of cells (Table S1)[9,12–14,17]. Recently, "single-cell" proteomics has been reported for proteome profiling of relatively large cells such as individual blastomeres isolated from *Xenopus laevis* embryos[18,19]. These measurements were enabled by the fact that each of these large cells contained micrograms of protein, compared to ~0.1 ng[20] of protein found in typical mammalian cells, and were thus compatible with conventional sample preparation protocols. Although <0.2% of the total digest (~20 ng tryptic peptides) from single blastomeres was injected for each analysis, an identification of 500–800 protein groups in single blastomeres was achieved and significant cell heterogeneity was found[18].

While progress has been made in enabling the proteomic analysis of small numbers of cells, a gap remains between required sample input and the demonstrated analytical sensitivity, and the robustness and reproducibility of most previous methods for biomedical applications have not yet been demonstrated. Innovation is required to further advance sample processing efficiency for nanoscale biological samples (i.e., containing nanogram or subnanogram amounts of protein) to enable deep, quantitative proteome profiling for such applications. Herein, we report a robotically addressed chip-based nanodroplet processing platform for enhancing proteomic sample processing and analysis for small cell populations. The platform, termed nanoPOTS (Nanodroplet Processing in One pot for Trace Samples), reduces total processing volumes from the conventional hundreds of microliters to <200 nL within a single droplet reactor. When coupled with highly sensitive LC-MS, we demonstrate that nanoPOTS enables reproducible and quantitative proteomic measurements of 1500–3000 protein groups (proteins) from 10 to 140 cells, a level of coverage only achieved previously for thousands of cells[9,12–16]. Further, we demonstrate the reproducible quantification of ~2400 proteins from single human pancreatic islet cross-sections isolated from 10-μm-thick pancreatic tissue slices, illustrating the enabling potential for molecular characterization of tissue cellular heterogeneity and pathology in type 1 or type 2 diabetes.

## Results

**NanoPOTS platform design and operation**. NanoPOTS glass chips were microfabricated with photolithographically patterned hydrophilic pedestals surrounded by a hydrophobic surface to serve as nanodroplet reaction vessels (nanowells) for multi-step proteomic sample processing. The chip consisted of the patterned glass slide (Supplementary Figure 1) and a glass spacer, which was sealed to a membrane-coated glass slide to minimize evaporation of the nanowell contents during the various incubation steps (Fig. 1a, b). The glass substrate facilitates microscopic imaging of samples and minimizes protein and peptide adsorption relative to many other materials due to its hydrophilicity and reduced surface charge at low pH[21]. The patterned nanoPOTS pedestals further reduce surface contact relative to the use of concave wells. A robotic platform[22] with submicron positioning accuracy and capacity for accurately handling picoliter volumes (Supplementary Figure 2) was used to dispense cells and reagents into nanodroplets and to retrieve samples for subsequent analysis. We adapted the RapiGest surfactant-based one-pot protocol for proteomic sample preparation with minimal modification (Fig. 1c). Briefly, after cells or other tissue samples were deposited into each chamber of the array, microscopic imaging was used for sample quantification (cell number, tissue dimensions, etc.). A cocktail containing RapiGest and dithiothreitol (DTT) was added and incubated at 70 °C to lyse cells, extract and denature proteins, and reduce disulfide bonds in a single step. The proteins were then alkylated and digested using a two-step enzymatic

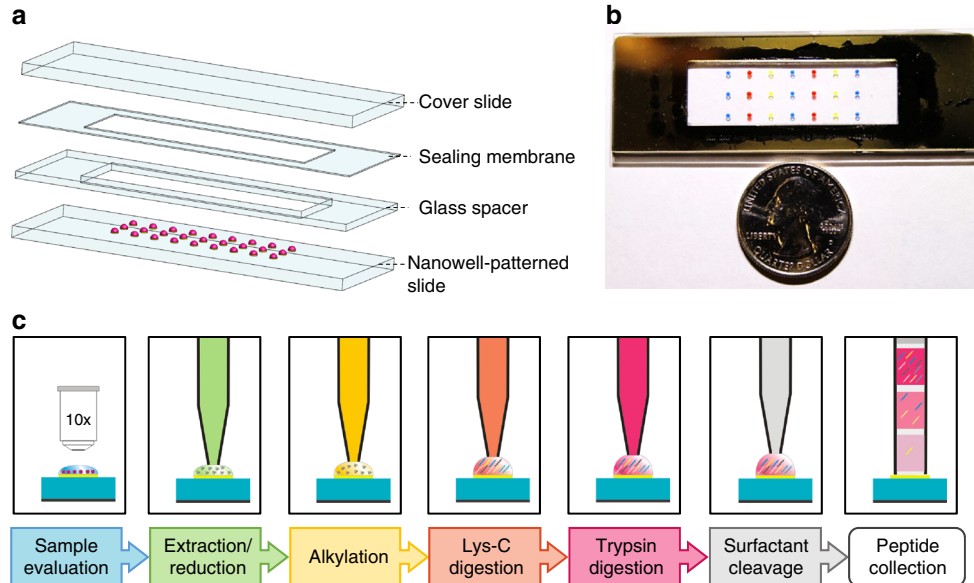

**Fig. 1** Proteomic sample preparation with nanoPOTS. **a** Schematic drawing and (**b**) photograph showing the nanoPOTS chip with each nanowell filled with 200 nL of colored dye. The cover slide can be removed and resealed for dispensing and incubation. **c** One-pot protocol for proteomic sample preparation and capillary-based sample collection

hydrolysis. Finally, the solution was acidified to cleave and inactivate the RapiGest surfactant. Manipulations were conducted in a humidified chamber, and the cover plate was sealed to the nanowell chip during extended incubation steps to minimize evaporation of the nanoliter droplets.

With the nanoPOTS platform, the entire processing procedure was performed within a 200 nL or smaller droplet that is contained in a wall-less glass reactor having a diameter of 1 mm (total surface area of ~0.8 mm$^2$). Compared with a typical sample preparation volume in a 0.5 mL centrifuge tube (~130 mm$^2$), the surface area was reduced by ~99.5%, greatly reducing adsorptive losses. Further, by preserving the ratio of protein to protease within the nanoPOTS platform that has been found to be optimal for bulk samples[23], the digestion rate is potentially increased significantly relative to a standard-volume preparation for the same number of cells.

The final processed sample was then collected into a fused silica capillary, followed by a two-step wash of the corresponding nanodroplet location to maximize recovery (Fig. 1c). The collection capillary can be fully sealed and stored in a freezer for months without observable sample loss. The capillary storage approach also simplified downstream solid-phase extraction (SPE)-based cleanup and LC-MS analysis by enabling direct coupling with standard fittings. Such a design can efficiently minimize sample losses during sample transfer and injection compared with autosampler-based injection methods.

**LC-MS platform for the ultrasmall samples**. To enable the analysis of the ultrasmall amounts of protein digest prepared using the nanoPOTS platform, the overall sensitivity of LC-MS is critical. We used 30-μm-i.d. nanoLC columns rather than the conventional 75-μm-i.d. columns, which substantially enhanced sensitivity due to increased ionization efficiency at the nanoelectrospray ion source and increased concentration of each component eluting from the narrow-bore columns[5,6,24]. Moreover, a state-of-the-art Orbitrap Fusion Lumos mass spectrometer was employed to maximize detection sensitivity and scan speed.

**Sensitivity and proteome coverage**. We first evaluated the sensitivity and achievable proteome coverage for processing and analyzing small numbers of cultured HeLa cells with nanoPOTS (Fig. 2a). Three different blank controls were used to confirm negligible carryover and contamination from the SPE and LC columns, reagents, and cell supernatant, respectively (Supplementary Figure 3). In contrast to the control samples, all cell-containing samples showed feature-rich base peak chromatogram profiles, and the number of peaks and their intensities increased with the number of cells (Fig. 2b and Supplementary Figure 4). The percentage of identified peptides having fully tryptic cleavage sites ranged from 97.4% to 97.9%, while the percentage of peptides with tryptic missed cleavage sites ranged from 23.2% to 27.8% (Supplementary Figure 5), indicating a digestion efficiency that is on par with conventional bulk processing[16]. The average MS/MS-based peptide identifications ranged from 7364 to 17,836, and protein identifications ranged from 1517 to 3056 for triplicate groups comprising 10–14, 37–45, and 137–141 cells, respectively (Fig. 2c, d). The average number of proteins was 965 to 2167 when at least two peptides were required for identification. When the Match Between Runs (MBR) algorithm of Maxquant[25] was used, average protein identifications increased to 3092, 3215, and 3460 for the smallest to largest cell loadings. Eighty-five percent of the identified proteins were found to be common to all samples (Supplementary Figure 6), indicating that more proteins could be identified and quantified from the smaller samples if a reference sample containing more cells was analyzed in parallel. When the proteins were constrained to contain at least two unique peptides, the number of proteins was 2356, 2509 and 2798 for the smallest to largest cell loadings, respectively. We further employed open-source quality control software to evaluate the quality of MBR identification[26]. For all datasets used in this study, both MBR alignment and ID-transfer metrics (Supplementary Figure 7) indicate high confidence of the MBR-transferred identifications. An independent evaluation of mass error distribution of MBR data points provided an estimated false discovery rate (FDR) of 2.03% (Supplementary Figure 8), again supporting the high confidence of MBR identifications[27].

The ability to identify an average of 3092 proteins from as few as ~10 cells (Fig. 2d) represents a >2 order of magnitude decrease in

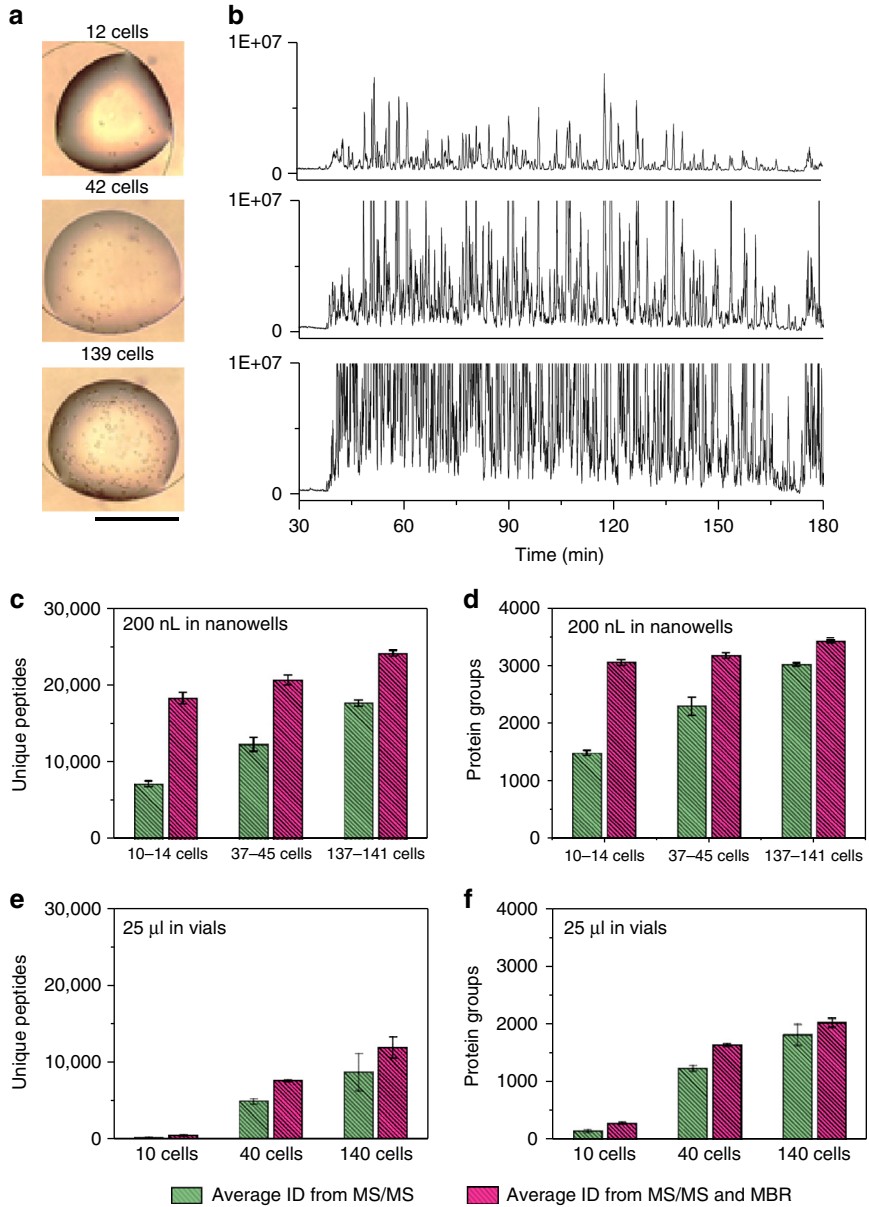

**Fig. 2** Evaluation of sensitivity and reproducibility for the nanoPOTS platform. **a** Images of 12, 42, and 139 HeLa cells in nanowells before processing and (**b**) their corresponding base peak chromatograms. The y-axis was held constant to show signal variation with different cell loadings. **c** Number of unique peptides and (**d**) proteins identified from different cell loadings. Three samples containing similar cell numbers were grouped together to determine reproducibility. For the 10–14 cells group, the cell numbers were 10, 12, and 14 cells. For 37–45 cells group, the cell numbers were 37, 42, and 45 cells. For 137–141 cells group, the cell numbers were 137, 139, and 141 cells. **e** Number of unique peptides and (**f**) proteins identified from cell lysate equivalent to 10, 40, and 140 cells. The cell lysates were prepared in regular Eppendorf low-binding vials with 25 μL of total processing volume. Data are expressed as means ± SD for experimental triplicates. Scale bar in (**a**) is 500 μm

sample amount for achieving similar proteome coverage relative to previously reported methods (Supplementary Table 1)[9,13–16]. Based on a quantitative study of HeLa cell growth during the culture cycle[28], the protein content in single HeLa cells was ~150 pg per cell. Thus, the estimated total protein contents in 10–14, 37–45, and 137–141 cells were ~1.5, 6, and 21 ng, respectively. This level of sensitivity for analyzing low-nanogram samples opens new possibilities in various applications involving limited sample supplies.

To further validate the contribution of nanoPOTS processing, we compared the performance of nanoliter versus larger volume proteomic sample processing using 10–140 HeLa cells with the same LC-MS conditions. The total dispensed volume in the regular Eppendorf low-binding vials was 25 μL, which has been used previously for highly sensitive proteomic sample

preparation[9]. The average peptide identifications ranged from 743 to 12,077, and protein identifications from 313 to 2048 for 10, 40, and 140 cells, respectively (Fig. 2e, f). We note that these vial-based processing results were still an improvement over what has been reported previously for small-scale proteomics (Supplementary Table 1), presumably due to improved LC-MS performance. However, these identifications are still significantly lower than those from nanoPOTS processing (Fig. 2c, d). In addition, the performance gain of nanoPOTS over vial-based preparation was much higher for the samples containing the fewest cells. A ~25-fold increase in peptide identifications was observed for ~10-cell samples, whereas only a 2-fold increase was realized for ~140-cell samples, which further demonstrates that the nanoPOTS platform is especially beneficial for ultrasmall samples.

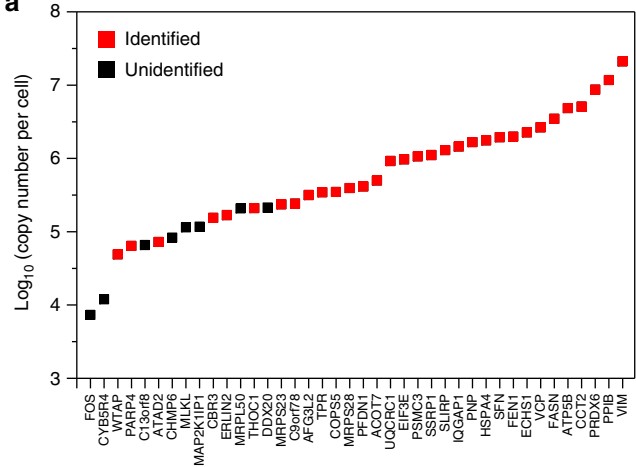

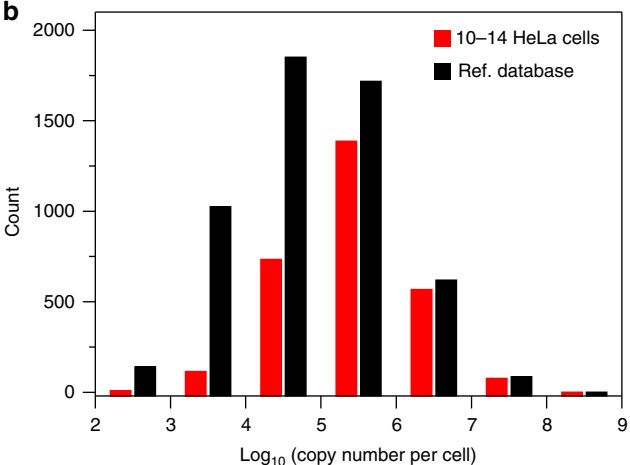

**Fig. 3** Distribution of protein abundances identified from 10 to 14 HeLa cells. **a** Identified proteins from among the 40 proteins quantified using the PrEST-SILAC method[30]. **b** Comparison with the 5443 proteins quantified using the histone-based "proteomic ruler" method. Protein copy numbers per cell were obtained by matching them with previously reported databases[20]

To assess the absolute sensitivity of the overall nanoPOTS-LC-MS platform, we matched the proteins identified from 10 to 14 cells to the reported databases containing protein copy numbers per cell for HeLa cells[20,29]. In the first database, the absolute copy numbers for 40 proteins in HeLa cells were precisely quantified using spiked-in protein epitope signature tags (PrEST) in combination with stable isotope labeling with amino acids in cell culture (SILAC)-based isotopic labeling[29]. We identified 32 of the 40 proteins, and the 8 missed proteins were all in the low-abundance range (Fig. 3a). The corresponding median values of protein copy number per cell ranged from $\sim 5 \times 10^4$ to $\sim 2 \times 10^7$ (Supplementary Table 2). Considering the highly reliable values obtained using the PrEST-SILAC method, we can confidently conclude the detection limit of nanoPOTS for protein is $< 5 \times 10^5$ copies, or <830 zmol. In the second database, a total of >5000 proteins in HeLa cells were quantified using a histone-based "proteomic ruler" and label-free quantification based on MS intensities[20]. Two thousand eight hundred and ninety-two of these proteins matched the proteins identified in our 10–14-cell samples, and the distribution of copy numbers per cell is shown in Fig. 3b. As expected, our results are somewhat biased to high-abundance proteins due to the use of only ~10 cells, and the median copy number within our samples was $\sim 2.5 \times 10^5$, which is

approximately four times higher than the reference value[20]. Importantly, we also identified a number of low-abundance proteins, including 125 proteins with copy numbers below $1 \times 10^4$ (Fig. 3b). These results indicate that the absolute detection limits of the nanoPOTS-LC-MS platform may be below 16 zmol in some cases.

**Reproducibility and quantification**. To assess whether the nanoPOTS platform can provide comparative quantitative proteome profiling, its reproducibility using MS[1] intensity measurements was assessed at both the peptide and protein levels using label-free quantification. MBR analysis produced 13,194 quantifiable peptides and 2674 proteins for 10–14 cells (Supplementary Figure 9). Pairwise analysis of any two samples with similar cell loadings showed Pearson's correlation coefficients from 0.91 to 0.94 (Supplementary Figure 10) at the peptide level. Protein label-free quantification (LFQ) intensity revealed excellent correlations, with coefficients of 0.98 to 0.99 (Fig. 4a–c and Supplementary Figure 11). Median coefficients of variation (CVs) were ≤20.4% (peptide level) and 13.1% (protein level) for all the three cell loading groups (Fig. 4d and Supplementary Figure 12). Peptide and protein intensities spanning ~4 orders of magnitude were observed (Supplementary Figure 13), indicating that dynamic range and depth of measurement are substantially retained relative to bulk analyses. The reproducibility in terms of correlation and CVs is similar or even better than other LFQ data from different platforms such as our recent simplified nanoproteomic platform[30], which was successfully applied to quantify biological differences in the lung cellular proteome at various stages of development. Together, these data suggest that robust label-free quantification is feasible for ~10–100 cells, a proteomic sample amount far smaller than has been previously accessible.

**Application to single human pancreatic islet sections**. To further demonstrate potential applications involving characterization of tissue substructures or molecular phenotyping of heterogeneous tissues such as human pancreas, we applied this method to analyze 10-µm-thick cross-sections of individual human islets (Fig. 5a) that were isolated by laser microdissection from clinical pancreatic tissue slices (Supplementary Figure 14). In this pilot study, 18 randomly selected single islet sections comprising nine islets from a non-diabetic control donor and nine islets from a type 1 diabetes (T1D) donor were analyzed. The islet equivalents (IEQs) were calculated to be from 0.06 to 0.44, corresponding to approximately 90 to 454 cells based on their volumes and a previous quantitative study[31] (Supplementary Table 3). An average of 2676 and a total of 3219 proteins were identified for the 18 single islet sections. The average number of proteins was 1934 when at least two peptides were required for identification. The number of protein identifications exceeded those of previously reported single intact islets[15]. Two thousand four hundred and twenty-one proteins were quantifiable with >3 valid LFQ intensity values in at least one experimental condition.

Pairwise correlation analysis of protein LFQ intensities of the nine islets from the control donor resulted in coefficients ranging from 0.93 to 0.97 (Fig. 5b). The data again suggest relatively good reproducibility for analyzing these single islet sections. Gene ontology analysis indicated that the proteome data provided coverage of cellular compartments similar to bulk analyses (Supplementary Figure 15), demonstrating that nanoPOTS does not bias protein extraction from different cellular compartments.

Importantly, significant differences in abundance were observed for 304 proteins between T1D and control islets (FDR <2%, Supplementary Data 1 and Supplementary Figure 16). Figure 5c further highlights the drastic alterations in the

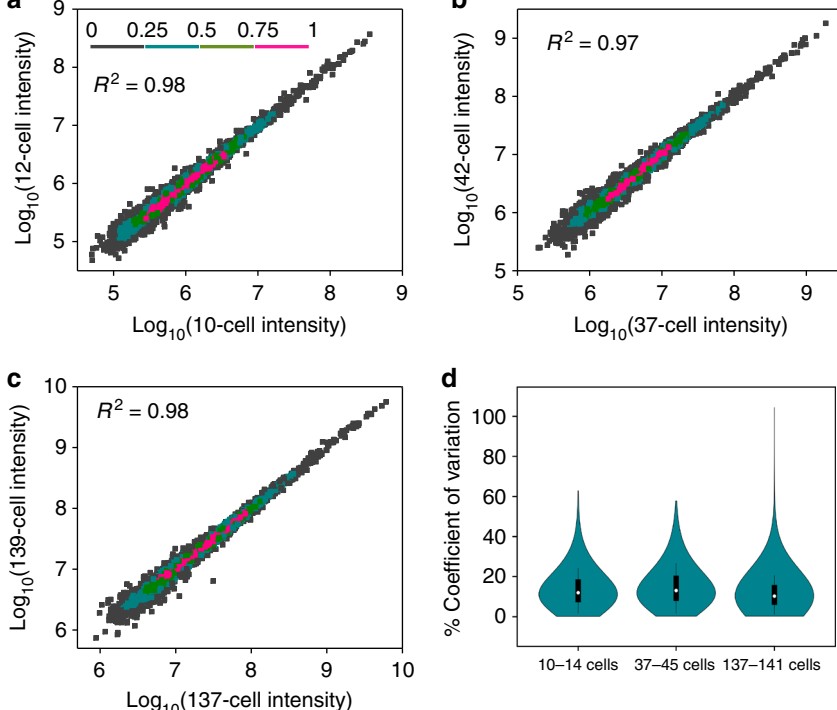

**Fig. 4** Label-free quantification reproducibility. Pairwise correlation of protein LFQ intensities between (**a**) 10-cell and 12-cell samples, (**b**) 37-cell and 42-cell samples, (**c**) 137-cell and 141-cell samples. Data point densities are color-coded as shown in (**a**). **d** Violin plot showing the distributions of coefficients of variation of protein LFQ intensities for the three cell loading groups. Center lines show the medians; box limits indicate the 25th and 75th percentiles as determined by R software; whiskers extend 1.5 times the interquartile range from the 25th and 75th percentiles

abundance of several proteins relevant to T1D pathology. First, glucagon (GCG), an alpha-cell-specific hormone, displayed similar abundances between T1D and control islets. GCG is an informative control marker to illustrate the robustness of single islet proteomics because alpha-cell masses are not significantly impacted in T1D. On the other hand, the significant reduction of beta-cell-specific markers of insulin and PCSK1 confirms a high degree of beta-cell loss in T1D islets. Another interesting observation is the increased expression of HLA-related protein products including beta-2-microglobulin since islet cell hyper-expression of HLA class I antigens have been reported as a defining feature of T1D[32,33]. Together, these data demonstrate that nanoPOTS is an enabling technology for studying pancreatic islets at the single islet level as a means of gaining insight into type 1 or type 2 diabetes pathology[34].

## Discussion

The nanoPOTS platform provides a robust, semi-automated nanodroplet-based proteomic processing system for handling extremely small biological samples down to as few as 10 cells with high processing efficiency and minimal sample loss. This capability opens many potential biomedical applications from small cell populations and clinical specimens such as tissue sections for characterizing tissue or cellular heterogeneity. Reproducible quantitative proteome measurements with coverage of 2000–3000 proteins from as few as 10 mammalian cells or single human islet cross-sections (~100 cells) from clinical specimens were demonstrated. While several previous efforts have pursued the analysis of <2000 cells, most of these methods lacked the robustness and reproducibility for biological applications because of the highly manual processes involved[9,13–15]. The nanoPOTS platform not only provides unparalleled proteome coverage for analyzing 10–100 cells, but also offers a number of technical advantages for

achieving a high degree of robustness and reproducibility for high-throughput processing and quantitative measurements when coupled with LC-MS. First, the platform effectively addresses the bottleneck of sample losses during proteomic sample preparation by performing all the multi-step reactions within a single nanodroplet of <200 nL volume, while all previous methods still suffer from a significant degree of protein/peptide losses during processing. Second, the nanodroplet processing mechanism allows us to perform each reaction at optimal concentrations. For example, by preserving both protein and protease concentrations within the nanodroplet without dramatic dilution, the digestion rate and efficiency is potentially significantly increased compared to a standard-volume preparation for the same number of cells[23]. Finally, in addition to label-free quantification, other stable isotope-based quantification methods should be readily adaptable to the workflow.

Compared with other microfluidic platforms having closed microchannels and chambers[35,36], nanoPOTS has an open structure, which is inherently suitable for integration with upstream and downstream proteomic workflows, including sample isolation for processing and transfer for LC-MS analysis. In addition to laser microdissection as demonstrated in this work, preliminary experiments have also shown the nanoPOTS chip can directly interface with fluorescence-activated cell sorting (FACS) for cell isolation. With the photolithography-based microfabrication technique, the nanodroplet array size and density can be easily scaled for increased preparation throughput.

While the currently demonstrated limit is as few as 10 cells, nanoPOTS represents a highly promising platform towards single mammalian cell proteomics with optimized processing volumes and further refinements to the analytical platform. To maximize the overall sensitivity of nanoPOTS for single cells, the total processing volume could be reduced to the low-nanoliter range to

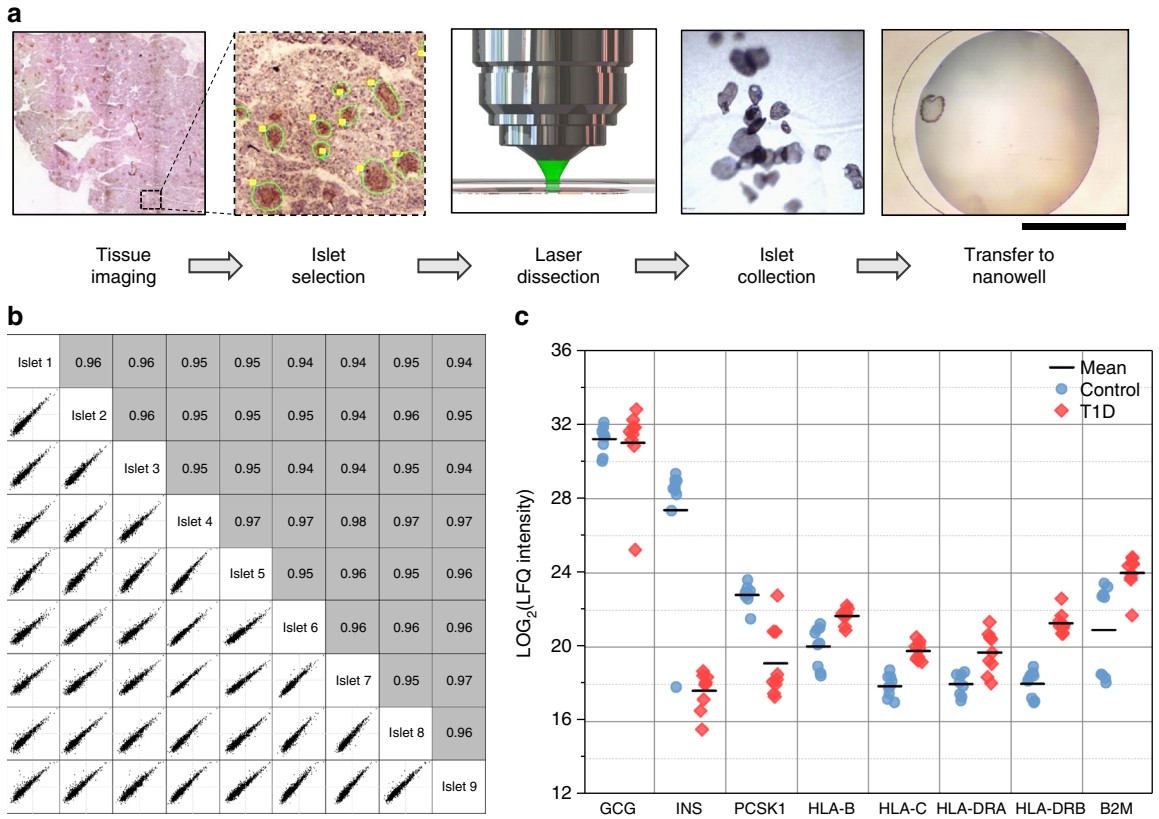

**Fig. 5** Proteomic profiling of single human pancreatic islet sections. **a** Schematic workflow showing the identification, laser microdissection, collection, and transfer of human islet sections into nanowells. **b** Pairwise correlation analysis of protein expression level in nine human islet sections from a non-diabetic donor. **c** Comparison of protein abundances in single islet sections between a non-diabetic donor (control) and a T1D donor. Scale bar is 500 μm

further minimize sample loss. FACS or other cell isolation techniques could be used to isolate single cells into nanowells without the minimal exogenous contamination from, for example, secreted proteins or lysed cells. NanoLC columns with narrower bore[6,24], and ESI emitter technology accommodating the lower resulting flow rates[5], could be employed to improve the detection sensitivity of the LC-MS system. Finally, in addition to single-cell analysis, nanoPOTS should also provide a viable path towards tissue imaging at the proteome level by performing in-depth spatially resolved proteome measurements for specific cellular regions.

## Methods

**Reagents and chemicals.** Deionized water (18.2 MΩ) was purified using a Barnstead Nanopure Infinity system (Los Angeles, CA, USA). DTT and iodoacetamide (IAA) were purchased from Thermo Scientific (St. Louis, MO, USA) and freshly prepared in 50 mM ammonium bicarbonate buffer each day before use. RapiGest SF surfactant (Waters, Milford, MA, USA) was dissolved in 50 mM ammonium bicarbonate buffer with a concentration of 0.2% (w/w), aliquoted, and stored at −20 °C until use. Trypsin (MS grade) and Lys-C (MS grade) were products of Promega (Madison, WI, USA). Other unmentioned reagents were obtained from Sigma-Aldrich (St. Louis, MO, USA).

**Fabrication and assembly of the nanowell chip.** The photomask was designed with AutoCAD and printed with a direct-write lithography system (SF-100, Intelligent Micro Patterning LLC, St. Petersburg, FL, USA). An array of 3 × 7 spots with diameters of 1 mm and a spacing of 4.5 mm was designed on a 25 mm × 75 mm glass slide (soda lime) that was pre-coated with chromium and photoresist (Telic Company, Valencia, CA, USA). After photoresist exposure (Supplementary Figure 1a), development, and chromium etching (Transene, Danvers, MA, USA; Supplementary Figure 1b), the glass slide was hard baked at 110 °C for 10 min. The back side of the slide was protected with packing tape and the glass surface was etched around the patterned photoresist/Cr features using wet etching solution containing 1 M HF, 0.5 M NH₄F, and 0.75 M HNO₃ at 40 °C for 10 min to reach a depth of 10 μm (Supplementary Figure 1c). The remaining photoresist was

removed using AZ 400T stripper. The glass slide was thoroughly rinsed with water, dried using compressed nitrogen, and further dried in an oven at 120 °C for 2 h. The chip surface was then cleaned and activated with oxygen plasma treatment for 3 min using a March Plasma Systems PX250 (Concord, NH, USA). The glass surface that was not protected with Cr was rendered hydrophobic with a solution containing 2% (v/v) heptadecafluoro-1,1,2,2-tetrahydrodecyl)dimethylchlorosilane (PFDS) in 2,2,4-trimethylpentane (Supplementary Figure 1d) for 30 min. The residual silane solution was removed by immersing the chip in 2,2,4-trimethylpentane followed by ethanol. Remaining chromium was removed using chromium etchant (Transene), leaving elevated hydrophilic nanowells on a hydrophobic background (Supplementary Figure 1e).

The glass spacer was fabricated by milling a standard microscope slide (25 mm × 75 mm × 1 mm) with a CNC machine (Minitech Machinery Corporation, Norcross, GA, USA). Epoxy was used to glue the patterned chip and the glass spacer together. The glass cover was fabricated by spin coating a thin layer of polydimethylsiloxane (PDMS) membrane (10-μm thickness) onto a standard glass microscope slide of the same dimensions. Briefly, Dow Corning Sylgard 184 silicone base was mixed with its curing reagent at a ratio of 10:1 (w/w) and degassed for 20 min. The mixture was coated on the slide by spinning at 500 rpm for 30 s, followed by 3000 rpm for 5 min (WS-650, Laurell Technologies, North Wales, PA, USA). Finally, the PDMS membrane was cured at 70 °C for 10 h. A piece of Parafilm (Bemis Company, Oshkosh, WI, USA) was precisely cut to serve as moisture barrier between the glass spacer and the glass cover.

**Nanoliter-scale liquid handling system.** All sample and reagent solutions were delivered to the nanowells using a home-built liquid handling system with a metering precision of 0.3 nL. The liquid handling system[22,37,38] was composed of four parts including a 3D translation stage (SKR series, THK, Tokyo, Japan) for automated position control, a home-built high-precision syringe pump (KR series, THK, Tokyo, Japan) for liquid metering, a microscopic camera system (MQ013MG-ON, XIMEA Corp., Lakewood, CA, USA) for monitoring the liquid handling process, and a tapered capillary probe for liquid dispensing. The capillary probe was fabricated by heating and pulling a fused silica capillary (200 μm i.d., 360 μm o.d., Polymicro Technologies, Phoenix, AZ, USA) to generate a tapered tip (30 μm i.d., 50 μm o.d.). A home-built program with LabView (Version 2015, National Instruments, Austin, TX, USA) was used to synchronously control the movement of the 3D stages and the liquid dispensing of the syringe pump. To

minimize evaporation during the liquid handling procedure, the whole system was enclosed in a Lexan chamber maintained at 95% relative humidity.

The syringe pump was set at a withdraw rate of 9 μL/min and an infusion rate of 3 μL/min. The translation stages were operated at a start speed of 1 cm/s, a maximum speed of 30 cm/s, and an acceleration time of 0.5 s. In the typical setup, it took total ~2 min to dispense one reagent to all the 21 droplets in single chip including the time for withdrawing reagent into the capillary probe, moving of the robotic stages, and dispensing 50 nL reagent into each droplet.

To meet the requirement of processing large number of samples in single experiment, the nanowells can be scaled up with the present photolithography-based microfabrication technique. Up to 350 nanowells can be fabricated on a 25 mm × 75 mm microscope slide and further scale-up is possible with larger substrates. The robot can be simply configured to fit different formats of nanowell array. Because of the high liquid handling speed, 350 droplets could be addressed in <30 min.

**Cell culture**. All cells were cultured at 37 °C and 5% $CO_2$ and split every 3 days following standard protocol. HeLa cells (ATCC) were grown in Eagle's minimum essential medium supplemented with 10% fetal bovine serum and 1× penicillin streptomycin.

**Laser microdissection of human pancreatic islets**. Ten-micrometer-thick pancreatic tissue slices from pancreata recovered from organ donors through the JDRF Network for Pancreatic Organ Donors with Diabetes (nPOD) program were cut from OCT (optimal cutting temperature compound) blocks using a cryo-microtome and mounted on PEN slides for islet dissection. Slides were briefly fixed with methanol, rinsed with $H_2O$ to remove OCT, and dehydrated using an alcohol gradient before placing in a desiccator to dry (8 min). Dehydrated and dried slides were placed on the stage of a laser microdissection microscope (Leica LMD7000). Islets were identified based on autofluorescence and morphology. Dissections were performed under a 10× objective. Laser dissected islets were collected in the cap of a 0.6 mL tube mounted underneath the slides. After dissection, samples were stored at −80 °C until further analysis. Ethical permission was obtained from the Institutional Review Boards at the University of Florida and Pacific Northwest National Laboratory, and informed consent was obtained from a legal representative of each donor.

**Proteomic sample preparation in Eppendorf low-binding vial**. HeLa cells were collected in a 10 mL tube and centrifuged at $171 \times g$ for 10 min to remove culture media. The cell pellet was further washed three times with 10 mL of 1× PBS buffer. The cells were then suspended in 1 mL PBS buffer and counted to obtain cell concentration. Eppendorf protein low-binding vials (0.5 mL) were used throughout the process. Cells were lysed at a concentration of $5 \times 10^5$/mL in 0.1% RapiGest and 5 mM DTT in 50 mM ammonium bicarbonate (ABC). After heating at 70 °C for 30 min, the cell lysate was diluted in 50 mM ABC buffer and aliquoted to different vials with a volume of 5 μL. Five microliters of IAA solution (30 mM in 50 mM ABC) was dispensed to alkylate sulfhydryl groups by incubating the vials in the dark for 30 min at room temperature. Five microliters of Lys-C (0.25 ng in 50 mM ABC) was added and incubated at 37 °C for 4 h. Five microliters of trypsin (0.25 ng in 50 mM ABC) was added and incubated overnight at 37 °C. Finally, 5 μL of formic acid solution (30%, v/v) were dispensed and allowed to incubate for 1 h at room temperature to cleave RapiGest surfactant for downstream analysis.

**Proteomic sample preparation in nanodroplets**. Before use, the chip was washed with isopropanol and water to minimize contamination. The liquid handling system was configured to minimize cross-contamination by adjusting the vertical distance between the probe tip and the nanowell surface, which was previously termed semi-contact dispensing[38].

For cultured cell samples, cells were collected in a 10 mL tube and centrifuged at $171 \times g$ for 10 min to remove culture media. The cell pellet was further washed three times with 10 mL of 1× PBS buffer. The cells were then suspended in 1 mL PBS buffer and counted to obtain cell concentration. Cell concentrations were adjusted by serially diluting them in PBS to obtain different cell numbers in nanowells. After dispensing 50 nL of cell suspension into each nanowell, we observed that the distribution of cell numbers in nanowells was stochastic, especially for low-concentration cell suspensions. Thus, the accurate cell number in each nanowell was counted using an inverted microscope and indexed to the two-dimensional spatial position of the corresponding nanowell. For LCM tissues, a high-precision tweezer with a tip of 20 μm (TerraUniversal, Buellton, CA, USA) was used to transfer tissue pieces from collection tubes into individual nanowells under a stereomicroscope (SMZ1270, Nikon, Tokyo, Japan). ImageJ software[39] was used to measure the area of LCM islets to calculate IEQ and cell numbers.

For sample preparation of cultured cells, 50-nL RapiGest[40] (0.2%) solution with 10 mM DTT in 50 mM ABC was added to the nanodroplets that had been preloaded with cells. For LCM tissue samples, 100 nL of RapiGest solution (0.1% in 50 mM ABC) containing 5 mM DTT was added. The cover was then sealed to the nanodroplet chip, which was incubated at 70 °C for 30 min to achieve cell lysis, protein denaturation, and disulfide reduction. In the second step, 50 nL of IAA solution (30 mM in 50 mM ABC) was dispensed to alkylate sulfhydryl groups by

incubating the chip in the dark for 30 min at room temperature. In the third step, 50 nL enzyme solution containing 0.25 ng Lys-C in 50 mM ABC was added and incubated at 37 °C for 4 h for predigestion. In the fourth step, 50 nL of enzyme solution containing 0.25 ng trypsin in 50 mM ABC was added to each droplet and incubated overnight at 37 °C for tryptic digestion. Finally, 50 nL of formic acid solution (30%, v/v) was dispensed and allowed to incubate for 1 h at room temperature to cleave RapiGest surfactant for downstream analysis. To minimize liquid evaporation in nanowells, the chip was completely sealed during cell counting, incubation, and transfer procedures. During each dispensing step, the chip was opened and closed within the humidity chamber to minimize droplet evaporation. However, as the total dispensed volume in each droplet was 300 nL and the final volume was typically <200 nL, some evaporative losses clearly occurred. Some of these water losses were observed as condensation on the contactless cover upon cooling from the 70 °C protein extraction step, and the extended digestions at 37 °C also resulted in minor volume reductions. Such water losses have no negative effect on the performance of nanoPOTS platform, but could become limiting when further downscaling processing volumes.

**Nanoliter-volume sample collection and storage**. Digested peptide samples in each nanowell were collected and stored in a section of fused silica capillary (5 cm long, 150 μm i.d., 360 μm o.d.). Before sample collection, the capillary was connected to the syringe pump and filled with water containing 0.1% formic acid (LC Buffer A) as carrier. A plug of air (10 nL, 0.5 mm in length) was aspirated into the capillary inlet to separate sample from carrier. The capillary-to-nanowell distance was adjusted to ~20 μm to allow the majority of sample to be aspirated into the capillary. To achieve highest sample recovery, the nanowell was twice washed with 200-nL Buffer A and the wash solutions were also collected in the same capillary. A section of capillary containing a train of plugs comprising carrier, air bubble, sample, and wash solutions was then cut from the syringe pump. The capillary section was sealed with parafilm at both ends and stored at −20 °C for short-term storage or −70 °C for long-term storage.

**SPE-LC-MS setup**. The SPE precolumn and LC column were slurry-packed with 3-μm C18 packing material (300-Å pore size, Phenomenex, Torrance, CA, USA)[6,24]. The SPE column was prepared from a 4-cm-long fused silica capillary (100 μm i.d., 360 μm o.d., Polymicro Technologies, Phoenix, AZ, USA). The LC column was prepared from a 70-cm-long Self-Pack PicoFrit column with an i.d. of 30 μm and a tip size of 10 μm (New Objective, Woburn, USA). The sample storage capillary was connected to the SPE column with a PEEK union (Valco instruments, Houston, USA). Sample was loaded and desalted in the SPE precolumn by infusing Buffer A (0.1% formic acid in water) at a flow rate of 500 nL/min for 20 min with an nanoACQUITY UPLC pump (Waters, Milford, USA). The SPE precolumn was reconnected to the LC column with a low-dead-volume PEEK union (Valco, Houston, USA). The LC separation flow rate was 60 nL/min, which was split from 400 nL/min with a nanoACQUITY UPLC pump (Waters, Milford, MA, USA). A linear 150-min gradient of 5–28% Buffer B (0.1% formic acid in acetonitrile) was used for separation. The LC column was washed by ramping Buffer B to 80% in 20 min, and finally re-equilibrated with Buffer A for another 20 min.

An Orbitrap Fusion Lumos Tribrid MS (ThermoFisher) was employed for all data collection. Electrospray voltage of 1.9 kV was applied at the source. The ion transfer tube was set at 150 °C for desolvation. S-lens radio frequency level was set at 30. A full MS scan range of 375–1575 and Orbitrap resolution of 120,000 (at $m/z$ 200) was used for all samples. The AGC target and maximum injection time were set at 1E6 and 246 ms. Data-dependent acquisition mode was used to trigger precursor isolation and sequencing. Precursor ions with charges of +2 to +7 were isolated with an $m/z$ window of 2 and fragmented by high energy dissociation with a collision energy of 28%. The signal intensity threshold was set at 6000. To minimize repeated sequencing, dynamic exclusion with duration of 90 s and mass tolerance of ±10 ppm was utilized. MS/MS scans were performed in the Orbitrap. The AGC target was fixed at 1E5. For different sample inputs, different scan resolutions and injection times were used to maximize sensitivity (240k and 502 ms for blank control and ~10-cell samples; 120k and 246 ms for ~40-cell samples; 60k and 118 ms for ~140-cell samples).

**Data analysis**. All raw files were processed using Maxquant (version 1.5.3.30) for feature detection, database searching, and protein/peptide quantification[25]. MS/MS spectra were searched against the UniProtKB/Swiss-Prot human database (downloaded in December 29, 2016 containing 20,129 reviewed sequences). N-terminal protein acetylation and methionine oxidation were selected as variable modifications. Carbamidomethylation of cysteine residues was set as a fixed modification. The peptide mass tolerances of the first search and main search (recalibrated) were <20 and 4.5 ppm, respectively. The match tolerance, de novo tolerance, and deisotoping tolerance for MS/MS search were 20, 10, and 7 ppm, respectively. The minimum peptide length was seven amino acids and maximum peptide mass was 4600 Da. The allowed missed cleavages for each peptide was 2. The second peptide search was activated to identify co-eluting and co-fragmented peptides from one MS/MS spectrum. Both peptides and proteins were filtered with a maximum FDR of 0.01. The MBR feature, with a match window of 0.7 min and an alignment window of 20 min, was activated to increase peptide/protein identification of low-cell-number samples. LFQ calculations were performed separately in each

parameter group containing similar cell loading. Both unique and razor peptides were selected for protein quantification. Requiring MS/MS for LFQ comparisons was not activated to increase the quantifiable proteins in low-cell-number samples. Other unmentioned parameters were the default settings of the Maxquant software.

Perseus software[41] was used to perform data analysis and extraction. The LFQ intensities were transformed using log 2 function. The extracted data were further processed and visualized with OriginLab 2017. Global scaling normalization was achieved using scaling coefficients calculated as the ratio of peptide abundance to the median peptide abundance measured for each loading set. CVs were defined as the standard deviation of normalized intensities divided by the mean intensity across the processing replicates of the same loading. The Violin plot was generated with an online tool (BoxPlotR, http://shiny.chemgrid.org/boxplotr/)[42].

To identify differentially expressed proteins between non-diabetic and T1D islets, the dataset was filtered to contain three valid values in at least one group. The missing values were imputed from normal distribution with a width of 0.3 and a down shift of 1.8. Two-sample $t$ test was applied with the Benjamini–Hochberg procedure[43] for FDR control. A $q$ value cutoff of 0.02 and a fold-change >2 were applied to identify proteins with significant abundance differences between T1D and control islets. The dataset was also analyzed with the same statistical procedure without imputation (Supplementary Data 1 and Supplementary Figure 16).

**Data availability**. The mass spectrometry proteomic data have been deposited to the ProteomeXchange Consortium via the PRIDE partner repository[44] with the dataset identifier PXD006847.

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

## Acknowledgements

We thank Michael S. Russcher, Randolph J. Norheim, and Daniel J. Orton for assisting with assembly of the robotic dispensing platform, Athena A. Schepmoes, and Tao Liu for donating HeLa cells, and Nikola Tolic for data processing. This work was supported by the NIH grants R21 EB020976 (to R.T.K.), R33 CA225248 (to R.T.K.), P41 GM103493 (to R.D.S.), UC4 DK104167 (to W.-J.Q.), and DP3 DK110844 (to W.-J.Q.). This research was performed using EMSL, a national scientific user facility sponsored by the Department of Energy's Office of Biological and Environmental Research and located at PNNL. This work utilized a LEICA 7000 laser microdissection microscope purchased with funding from a National Institutes of Health SIG grant 1S10OD016350-01. This research

was also performed with the support of nPOD, a collaborative type 1 diabetes research project sponsored by JDRF. Organ Procurement Organizations (OPO) partnering with nPOD to provide research resources are listed at http://www.jdrfnpod.org/for-partners/npod-partners/.

## Author contributions

Y.Z., W.-J. Q., and R.T.K. proposed the method and designed the research. Y.Z. and R.T.K. designed and fabricated the nanowell chip. Y.Z. designed and assembled the robotic dispensing platform and performed proteomic sample preparation experiments. R.Z. and Y.S. prepared the LC and SPE columns. R.J.M. and A.K.S. optimized the MS system. Y.Z. conducted LC-MS experiments. J.C., M.C.-T., and C.E.M. performed laser microdissection of single human islets. Y.Z., P.D.P., V.A.P., W.-J.Q., and R.T.K. analyzed the data. Y.Z., P.D.P., R.D.S., W.-J.Q., and R.T.K. wrote the manuscript.

## Additional information

**Competing interests:** The authors declare no competing interests.

