## [Peer Review File · Nature Communications]

Reviewers' comments:

Reviewer #1 (Remarks to the Author):

Reducing sample losses is one of the essential steps required for improving our ability to quantify proteins in small samples. Working towards this step, Zhu et al. developed a sample preparation platform that they call (Nanodroplet Processing in One pot for Trace Samples) NanoPOTS. The key innovation is the ability to reduce the volume of a sample to below 200 nl and thus minimize losses. The authors reasoned that reduced volumes would reduce losses and thus should enhance the sensitivity for samples comprised of tens or hundreds of cells.

The NanoPOTS platform consists of microfabricated glass chips and robotic arms. This appears to be a carefully constructed system that is indeed able to do sample preparation in very small volumes. I am curious to know what is the throughput of the system, its robustness and potential for scaling up.

Despite the small volume of sample prep, the benefits for peptide and protein identification appear relatively modest. Previously, Li et al., *Molecular & Cellular Proteomics*, 2015 reported quantifying > 2000 proteins from 100 cells and over 1300 proteins from 50 cells (Figure 3 from Li et al). This is quite close to what Zhu et al. report in Fig. 2 without the matching between the runs. I noticed that the coverage of Li et al., *Molecular & Cellular Proteomics*, 2015 is misrepresented in Table S1, bottom row. Why not include the coverage that Li et al. (ref. 5) achieved with lower input material ?

To summarize: The NanoPOTS platform appears well built but it is not clear that it advances very significantly previous results, e.g., Li et al., *Molecular & Cellular Proteomics*, 2015. It is also unclear how flexible and generalizable the platform is.

I found the benchmarking of quantification unconvincing. This is not to say that data produced from the NanoPOTS platform is not quantitative but its quantitative accuracy is hard to assess from the figures in the paper. Matching the distributions in Fig. 3 is a very weak test. At least the authors could have presented the results as a scatter plot, plotting their estimates of protein abundances against those from the literature. Fig. 4 does provide some evidence of reproducibility, but it is hard to see what is the reproducibility or the accuracy of quantifying how a protein changes across samples. Is the quantification good enough to detect a 5 fold

change of a protein between 2 samples ? I cannot tell from that figure.

The authors apply their method to profile heterogeneity in the pancreas but judge the quality of their data as "The data suggests good reproducibility for analyzing some of the single islet sections; however, slightly lower coefficients were observed between some other pair-wise comparisons, presumably due to the degree of islet heterogeneity from the non-diabetic subject". I see a very tight distribution of pairwise correlations between samples (Fig. 5b): 0.95 ± 0.02 . There are no clusters of islet indicating biological differences between islets or proteins identified as heterogeneously / differentially expressed. The only interpretation that I take away from these data is that the authors were able to detect peptides from thin slices of pancreas and the overall peptide/protein intensities did not differ much between islets.

Reviewer #2 (Remarks to the Author):

The paper describes novel bottom-up proteomic technique for sample preparation. The presented results suggest that this approach has >100 fold higher sensitivity compared to commonly used methods. It is unbelievable that analysis a lysate from 10 cells, which corresponds to about 2 ng total protein, gives a MS signal of a strength that in other laboratories is achievable when much higher amounts of digest are analyzed.

My essential question is how the high sensitivity of the mass spectrometer that authors use can be achieved? It would be beneficial for the proteomic community to learn it. Because data showing the yields of protein to peptide conversion are missing it is not possible to directly judge on the sample preparation method.

Reviewer #3 (Remarks to the Author):

In this work, a nanoPOTS platform was developed. Nanoscale droplets containing 10-100 cells were processed in a nanowell-patterned slide. Cell number was counted and then extraction, alkylation, proteolytic digestion, desalts were carried out in the droplet. Afterwards, the proteome sample was transferred into a nano-LC-MS system for proteomic analysis. About 2000~3000 proteins were identified for 10 to 100 cell samples. Such a method was applied to a sample analysis of single pancreatic islet proteome. After laser microdissection of single islets, around 2300 proteins in a single islet sample could be identified. It is proven such method was very powerful in 10-100 cell sample proteome analysis. Although such a method was involved too many operating steps and taken 12-24 hours for carrying on an analysis, it is still a very potential platform to an extremely small bio-sample analysis. Some questions need to be

additionally discussed:

- (1) how to control cell numbers in the droplets? Is it just randomly drawn into droplet?
- (2) The total volume of the droplets was 300 nL instead of <200 nL after added all reagents. Was part of the solutions vaporized during processing? Was it necessarily open-and-close the slide in each step?
- (3) Did the authors try a single cell proteome analysis using the nanoPOTS? What are the main difficulties to do this?
- (4) Proteome analysis for a single pancreatic islet is very attractive. Could the islet tissue samples be completely dissolved in buffer as that did same to the cultured cell samples?
- (5) How about the conclusion for features of heterogeneity to different pancreatic islets?

Reviewer #4 (Remarks to the Author):

Kelly et al. report an exciting technological development to advance the sensitivity of bottom-up proteomics to small populations of mammalian cells. As also expressed in the manuscript, this development is important, because the transcriptome does not (necessarily) describe the phenotype of cells, cell populations, or tissue, particularly in developing system. The functionally important proteins do. However, detection of proteins in such limited amounts of samples is demands exceptional sensitivity and is subject to non-specific protein losses on vial surfaces, pipettes, etc. This work directly addresses the issue of protein losses on vial surfaces.

The new instrument that was developed in this work, termed Nanodroplet Processing in One pot for Trace Samples, or simply nanoPOTS, uses precision robotics to handle samples and reagents and microfabricated glass chips to reduce the volume of bottom-up proteomics sample processing from the typical ~100 uL to just 200 nL. Furthermore, these microchips feature designs to reduce protein adsorption, leading to sensitivity improvements. The nanoPOTS system allowed nearly all steps of the bottom-up workflow to be carried out, ranging from protein extractions, lysis, reduction, to alkylation in 1000-times less volumes. Loading of the resulting samples into closed fused silica capillaries with potential freezing of the sample was an excellent way to transfer the resulting mass-limited samples without losing proteins in this work. Coupling of the peptide digest-containing capillaries to a separation system (UPLC) followed by high-resolution mass spectrometry detection conferred excellent sensitivity. For example, ~3,000 proteins were detected and quantified over 3-log-order dynamic range with excellent quantitative reproducibility (Pearson products ~ 0.9) in just ~10 or so HeLa cells in this work. This is a record, in my view! The protein identifications were carried out with 1% FDR and 2 peptides/protein, upholding high standards set by the proteomics community. The quantified proteins and their dynamic ranges agree with those available for large-population cells, which is assuring. In addition, this study already demonstrated applicability of the tool to laser capture

microdissection sampling of cells in single human pancreatic islets. These data demonstrate the usefulness of this technology not only for small cell-population analysis but also limited protein measurements.

In summary, I find the nanoPOTS system a novel development that raises important new possibilities for mass spectrometry-based proteomics. The technology is highly adaptable to existing mass spectrometry technology, and if commercialized (which I hope the researchers are already exploring), can be readily integrated into current proteomics workflows in many laboratories worldwide. The manuscript is well written using language with road audiences in mind.

I enthusiastically support publication of this work in Nature Communications. Additionally, I would like to make some suggestions and questions to the authors to help further improve this manuscript for general readers (non-experts in MS or bottom-up proteomics) – see below. I congratulate the team on this exciting technological development and look forward to seeing its success in the field.

Peter Nemes

(I, Peter Nemes, have authorized the release of my name on this peer review.)

COMMENTS/SUGGESTIONS

1/ In the Introduction and later throughout the manuscript, it would be helpful to remind readers that – unlike in transcriptomics – bottom-up proteomics does not (yet) benefit from whole-proteome amplification. Thus, minimizing sample losses is critical to the success of proteomics, which is exactly what nanoPOTS addresses.

2/ Introduction: I appreciate the authors' reference to our single-cell proteomics work in *Xenopus* (Ref. 18). While, indeed, *Xenopus* cells are considerably large, there is an important technical detail that I would appreciate highlighted to note the significance of our study for single-cell analysis. Namely, to demonstrate the utility of our single-cell CE-ESI-MS technology to mammalian cells, Ref. 18 identified ~1,700 proteins among cells (~800 proteins/cell) by analyzing not the whole *Xenopus* cell's proteome, but only a ~20 ng total protein content of the single cell, which is ~0.2% of cell's total protein amount. Of course, this is very different than analyzing the total protein content of whole dissected single cells from *Xenopus* (Ref. 19).

3/ The Slavov group recently deposited a manuscript on SCoPE-MS via bioRxiv (<https://doi.org/10.1101/102681>, title: "Mass-spectrometry of single mammalian cells quantifies proteome heterogeneity during cell differentiation"). I personally have not seen this work peer-reviewed yet, but if it has been published in a peer-reviewed format, then it could be a useful reference for this study.

4/ To demonstrate the sensitivity improvement in this work, it would be helpful to mention the total protein amount contained by the HeLa cells (~300 pg/cell, see ThermoFisher website) and

also the amount of total peptide/protein that was successfully extracted from the cells and then analyzed by the mass spectrometer in the case of small populations of HeLa cells. Perhaps, total peptide assay could be used for this purpose. How does this protein/peptide amount compare to those analyzed in single cells in the literature?

5/ The islet experiments used 2 unique peptides/protein for protein identifications. Was the same setting used (2 peptides/protein) for the limited-population cell experiments?

6/ Likewise, while the manuscript mentions: "Other unmentioned parameters were the default settings of the Maxquant software." To be comprehensive and avoid potential MaxQuant-version or user-installation dependent settings in MaxQuant, please spell out the search parameters (e.g., MS1 and MS2 mass accuracy, number of allowed missed cleavages, etc.).

7/ I have no doubts that the nanoPOTS system greatly minimizes protein/peptide losses during sample processing. However, one analytical question is how much role use of the new-generation mass spectrometer (Lumos) played in improving protein identifications and quantification. Based on presentations by Thermo, the Lumos improves sensitivity, charge-state identification, lower peptide redundancy, faster MSn events, etc. to boost identifications over earlier-generation mass spectrometers, which were also used in previous single-cell proteomics studies. So, to help demonstrate the improvement in relative terms, I would like to suggest a short follow-up experiment, namely to deposit the 10 or so cells into a regular Eppendorf vial and then carry out a standard bottom-up proteomics workflow on these cells using ~100 uL total volume (for reduction, alkylation, digestion), followed by analysis of the resulting sample using the same Lumos instrument under the same experimental settings that was used in this work. This does not take much time (few days max), but it would give a relative improvement in terms of protein identifications. In turn, this would help readers appreciate the significant sensitivity improvement by nanoPOTS vs. the standard approach, which is challenged by protein losses on vial surfaces, pipettes, etc.

Point-by-Point Response to the Reviewer Comments

Reviewer comments in italics, responses in blue.

Reviewer: #1

Reducing sample losses is one of the essential steps required for improving our ability to quantify proteins in small samples. Working towards this step, Zhu et al. developed a sample preparation platform that they call (Nanodroplet Processing in One pot for Trace Samples) NanoPOTS. The key innovation is the ability to reduce the volume of a sample to below 200 nl and thus minimize losses. The authors reasoned that reduced volumes would reduce losses and thus should enhance the sensitivity for samples comprised of tens or hundreds of cells.

The NanoPOTS platform consists of microfabricated glass chips and robotic arms. This appears to be a carefully constructed system that is indeed able to do sample preparation in very small volumes. I am curious to know what is the throughput of the system, its robustness and potential for scaling up.

We sincerely appreciate the Reviewer for the recognition to the novelty of our work, and we agree that this technique will fill the gap of current proteomic workflow for ultra-small amounts of biomaterials. The throughput of the nanoPOTS was determined by the moving speed of the translation stages and the flow rate of the syringe pump. The scaling up of the nanoPOTS is very straightforward with photolithography-based microfabrication technique. Both throughput and scaling up is not limiting at present considering the relatively long LC/MS analysis times required for proteomics studies. To address this, we added to following, beginning at the bottom of. 18 in the marked manuscript: “The syringe pump was set at a withdraw rate of 9 μ L/min and an infusion rate of 3 μ L/min. The translation stages were operated at a start speed of 1 cm/s, a maximum speed of 30 cm/s, and an acceleration time of 0.5 s. In the typical setup, it took \sim 2 min to dispense one reagent to all 21 droplets in a single chip including the time for withdrawing reagent into the capillary probe, moving the robotic stages, and dispensing 50 nL of reagent into each droplet.

To meet the requirement of processing large number of samples in single experiment, the nanowells can be scaled up with the present photolithography-based microfabrication technique. Up to 350 nanowells can be fabricated on a 25 mm \times 75 mm microscope slide and further scale-up is possible with larger substrates. The robot can be simply configured to fit different nanowell array formats. Because of the high liquid handling speed, 350 droplets could be addressed in <30 min.”

The robustness of the robotic system was well demonstrated in previous work since it was first developed in 2013 (the only substantial modification of the present system was the humidity chamber to minimize evaporation). It has been applied to various applications including enzyme inhibition assays (*Analytical Chemistry*, 2013, 85(14): 6723; *Analyst*, 2014, 139(1), 191; *Analytical Chemistry*, 2014, 86 (21),10796), protein crystallization screening (*Scientific Reports*, 2014, 4: 5046), digital PCR (*Analytical Chemistry*, 2017, 89 (1), 822), and single cell analysis

(*Scientific Reports*, 2015, 9551). For the robotic system dedicated for nanoPOTS, it typically took ~5 hours to train PhD students or research scientists to use it independently.

Despite the small volume of sample prep, the benefits for peptide and protein identification appear relatively modest. Previously, Li et al., Molecular & Cellular Proteomics, 2015 reported quantifying > 2000 proteins from 100 cells and over 1300 proteins from 50 cells (Figure 3 from Li et al). This is quite close to what Zhu et al. report in Fig. 2 without the matching between the runs. I noticed that the coverage of Li et al., Molecular & Cellular Proteomics, 2015 is misrepresented in Table S1, bottom row. Why not include the coverage that Li et al. (ref. 5) achieved with lower input material ?

We thank the Reviewer for pointing out this important point. The sensitivity achieved by Li et al. (*Mol. Cell. Proteomics* 14, 1672–1683 (2015)) is indeed impressive and is cited in our manuscript as ref. 9. However, Figure 3c in their manuscript is referring to analysis of aliquots of cell equivalents that are prepared in bulk from 10 million cells (p. 1675) and not actually starting with 50 or 100 cells. This is an important difference because the loss in sample processing was not considered in this portion of Li et al.' work. The smallest number of cells prepared in one tube was 1000 cells in Li et al., which was shown in the Fig. 4G and the page 1679. An average of 2512 proteins were identified from an aliquoted equivalent of 122 cells. When 2000 cells were processed (page 1677 of the paper and page 2 of its supplemental information), 3370 proteins from an aliquot of 500 cells were identified.

In Table S1, we aimed to summarize the protein identification starting with cell numbers lower than 2000, but not those with bulk sample preparation of large number cells followed by injection equivalent to small number cells. The later works such as Li et al. are excellent examples to show the high sensitivity of current LC-MS systems, but do not reflect the current state of the whole workflow for small samples. Thus, we cited the work by Li et al., and listed their result of 2512 and 3370 proteins from 1000 and 2000 MCF-7 cells in Table S1.

In the revised manuscript, to make a better definition of the protein identifications in Table S1, we added more explanations on the table caption as “Table S1. Previously published protein identification results starting with 2000 or less mammalian cells. Reports in which samples were prepared in bulk and subsequently analyzed with small number of cell equivalents were not included in this table.” on page 2 of Supporting Information.

To summarize: The NanoPOTS platform appears well built but it is not clear that it advances very significantly previous results, e.g., Li et al., Molecular & Cellular Proteomics, 2015. It is also unclear how flexible and generalizable the platform is.

As noted in our response above, nanoPOTS enables the processing of as few as 10 cells instead of processing 1000 or more cells as reported in Li et al. (*Mol. Cell. Proteomics* 14, 1672–1683 (2015)). Indeed, we did not claim any major advance in LC-MS, but present NanoPOTS as a novel processing system. The enabling capability to process 10 cells instead of ≥ 1000 cells is a significant advance. This advance is further demonstrated by a head-to-head comparison between vial-based processing and nanoPOTS (Page 8) as requested by Reviewer 4. “To further

validate the contribution of nanoPOTS processing, we compared the performance of nanoliter versus larger volume proteomic sample processing using 10–140 HeLa cells with the same LC-MS conditions. The total dispensed volume in the regular Eppendorf low-binding vials was 25 μ L, which has been used previously for highly sensitive proteomic sample preparation.⁹ The average peptide identifications ranged from 743 to 12,077, and protein identifications from 313 to 2,048 for 10, 40 and 140 cells, respectively (Fig. 2e, and 2f). We note that these regular vial-based processing results were still an improvement over what has been reported previously for small-scale proteomics (Table S1), presumably due to improved LC-MS performance. However, these identifications are still significantly lower than those from nanoPOTS processing (Fig. 2c, and 2d). In addition, the performance gain of nanoPOTS over vial-based preparation was much higher for the samples containing the fewest cells. A ~25-fold increase in peptide identifications was observed for ~10-cell samples, whereas only a 2-fold increase was realized for ~140-cell samples, which further demonstrates that the nanoPOTS platform is especially beneficial for ultrasmall samples.”

As we mentioned earlier in our response, the platform is fully flexible and generalizable in terms of robustness, automation, and scaling up.

I found the benchmarking of quantification unconvincing. This is not to say that data produced from the NanoPOTS platform is not quantitative but its quantitative accuracy is hard to assess from the figures in the paper. Matching the distributions in Fig. 3 is a very weak test. At least the authors could have presented the results as a scatter plot, plotting their estimates of protein abundances against those from the literature. Fig. 4 does provide some evidence of reproducibility, but it is hard to see what is the reproducibility or the accuracy of quantifying how a protein changes across samples. Is the quantification good enough to detect a 5 fold change of a protein between 2 samples? I cannot tell from that figure.

We emphasize that the Figure 3 is intended to show the absolute sensitivity of the overall nanoPOTS-LC-MS platform, not to assess the quantitative accuracy. Because both the previous work (ref. 20 and 25) reported the protein copy number per cell for HeLa, they are good resource to estimate how deep this nanoPOTS platform can detect when starting with extremely small samples (e.g., 10 HeLa cells). As we show in the manuscript, the reproducibility and quantitative accuracy was evaluated by pairwise analysis of peptide and protein LFQ intensities with similar cell loadings. The correlation of protein LFQ intensities as well as the coefficients of variations (<20% for protein LFQ) were similar or even slightly better than that of our recently our recently simplified nanoproteomics platform (SNaPP) (Clair, G. et al. Sci. Rep. 2016, 6, 39223).

To further address this, we have performed a pilot comparison of single islet sections from a non-diabetic donor and a T1D donor. We observed significant differences between T1D versus control islets (see Suppl. Fig 12 of volcano plot). Interesting abundance changes for selected T1D relevant proteins were highlighted in Figure 5c along with the following text (Page 13): “Fig. 5c further highlights the drastic alterations in the abundance of several proteins relevant to T1D pathology. First, glucagon (GCG), an alpha-cell specific hormone, displayed similar abundances between T1D and control islets. GCG serves an interesting control marker to

illustrate the robustness of single islet proteomics because alpha cell masses are not significantly impacted in T1D. On the other hand, the significant reduction of beta-cell specific markers of insulin and SCG5 confirms a high-degree of beta cell loss in T1D islets. Another interesting observation is the increased expression of HLA related protein products including beta-2-microglobulin (B2M) since islet cell hyperexpression of HLA class I antigens have been reported as a defining feature of T1D.”

The authors apply their method to profile heterogeneity in the pancreas but judge the quality of their data as "The data suggests good reproducibility for analyzing some of the single islet sections; however, slightly lower coefficients were observed between some other pair-wise comparisons, presumably due to the degree of islet heterogeneity from the non-diabetic subject". I see a very tight distribution of pairwise correlations between samples (Fig. 5b): 0.95 +/- 0.02. There are no clusters of islet indicating biological differences between islets or proteins identified as heterogeneously / differentially expressed. The only interpretation that I take away from these data is that the authors were able to detect peptides from thin slices of pancreas and the overall peptide/protein intensities did not differ much between islets.

The reviewer is correct in that our originally presented work did not identify specific information related to islet heterogeneity. However, we have included new results comparing T1D and control islets (Figure 5c). Abundance changes along with some evidence of heterogeneity were observed.

Reviewer #2:

The paper describes novel bottom-up proteomic technique for sample preparation. The presented results suggest that this approach has >100 fold higher sensitivity compared to commonly used methods. It is unbelievable that analysis a lysate from 10 cells, which corresponds to about 2 ng total protein, gives a MS signal of a strength that in other laboratories is achievable when much higher amounts of digest are analyzed. My essential question is how the high sensitivity of the mass spectrometer that authors use can be achieved? It would be beneficial for the proteomic community to learn it.

We thank the reviewer for raising this question. We believe our overall platform is fully translatable since the LC-MS component is based on commercially available instrumentation and the nanoPOTS platform can be disseminated through commercialization (we are currently working on such opportunity of dissemination). Besides the major contribution of the near lossless sample processing of nanoPOTS platform, the sensitivity of LC-MS must be optimized to enable the analysis of the ultra-small samples generated from nanoPOTS. We have clarified this point in the revised manuscript by adding the parameters of LC-MS conditions in the following paragraph (Page 7 of marked manuscript): “To enable the analysis of the ultra-small amounts of protein digest prepared using the nanoPOTS platform, the overall sensitivity of LC-MS is critical. We used 30- μ m-i.d. nanoLC columns rather than the conventional 75- μ m-i.d. columns, which substantially enhanced sensitivity due to increased ionization efficiency at the nanoelectrospray ion source and increased concentration of each component eluting from the

narrow-bore columns.^{5,6,24} Moreover, a state-of-the-art Orbitrap Fusion Lumos mass spectrometer was employed to maximize detection sensitivity and scan speed.”

Because data showing the yields of protein to peptide conversion are missing it is not possible to directly judge on the sample preparation method.

We agree that it will be beneficial to measure the yield of protein and peptide to directly evaluate the efficiency of sample preparation method. Unfortunately, current methods to measure protein and peptide yield are not applicable to low nanogram samples. However, the proteomic coverage results can give strong evidence to show the high efficiency of the nanoPOTS for protein extraction and digestion. First, the percentage of all the identified peptides having tryptic cleavage sites ranged from 97.4% to 97.9%, while the percentage of peptides with tryptic missed cleavage sites ranged from 23.2% to 27.8% (Fig. S4), indicating a digestion efficiency that is on par with conventional bulk processing (see Ref 16 in manuscript). Second, the identification of 3092 proteins in ~10 cells represents a 2–3 order of magnitude decrease in sample size to achieve similar proteome coverage relative to previously reported methods (see Refs 9, 13–16 in manuscript). Such performance indicates the total efficiency of nanoPOTS for protein extraction and digestion is much higher than other reported techniques.

5. Some more information is required for Figure 2. What was the concentration for each signal? Were there triplicates for each concentration? Some discussions are recommended for the reproducibility of this experiment (e.g. RSD).

We appreciate the Reviewer for the suggestions. In Figure 2b, the base peak chromatograms are from 12, 42, 139 HeLa cells as shown in the images of Figure 2a. The corresponding concentrations were not available due to the lack of peptide measurement technique for ultra-small samples as mentioned above. In Figure 2c and 2d, the error bars were generated from three samples containing similar numbers of HeLa cells. For example, for 10–14 cells, the cell numbers were 10, 12, and 14. As the suggestion of the Reviewer, in the revised manuscript, we added more details in the caption of Figure 2 (Page 9 of marked manuscript) as “Three samples containing similar cell numbers were grouped together to determine reproducibility. For the 10–14 cells group, the cell numbers were 10, 12, and 14 cells. For 37–45 cells group, the cell numbers were 37, 42, and 45 cells. For 137–141 cells group, the cell numbers were 137, 139, and 141 cells.”

The reproducibility of the experiment was well discussed in the section of **Reproducibility and quantification**, and the data were shown in Figure 4, Figure S7, Figure S8, and Figure S9. At the peptide level, the Pearson correlation coefficients were from 0.91 to 0.93 by pairwise analysis of any two samples with similar cell loadings (Figure S7). The median coefficients of variance (CVs or RSDs) were $\leq 20.4\%$ for the three cell loading groups (Figure S9). At the protein level, protein LFQ intensity revealed excellent correlations with coefficients of 0.98 to 0.99 (Fig. 4a–4c and Fig. S8). The median coefficients of variance (CVs or RSDs) were $\leq 13.1\%$ (Figure S9). These data revealed that the reproducibility of the nanoPOTS-LC-MS system was comparable or even better than other platforms for label-free quantification.

Reviewer #3

In this work, a nanoPOTS platform was developed. Nanoscale droplets containing 10-100 cells were processed in a nanowell-patterned slide. Cell number was counted and then extraction, alkylation, proteolytic digestion, desalts were carried out in the droplet. Afterwards, the proteome sample was transferred into a nano-LC-MS system for proteomic analysis. About 2000~3000 proteins were identified for 10 to 100 cell samples. Such a method was applied to a sample analysis of single pancreatic islet proteome. After laser microdissection of single islets, around 2300 proteins in a single islet sample could be identified. It is proven such method was very powerful in 10-100 cell sample proteome analysis. Although such a method was involved too many operating steps and taken 12-24 hours for carrying on an analysis, it is still a very potential platform to an extremely small bio-sample analysis.

We appreciate the Reviewer for the recognition to the novelty and the importance of our work. At present, we have adapted the standard RapiGest-based protocol, which was demonstrated to have strong performance for standard large sample amounts. This protocol typically took 12–24 hours because of the use of 4-hour Lys-c digestion and overnight trypsin digestion. However, we want to point out that large number of samples can be processed together to increase the throughput. For example, in our experiments, we commonly processed 42 samples in two chips per day. The throughput can be simply increased to hundreds even thousands of samples per day with the increase of nanowell number using photolithography-based microfabrication technique. Because of the high liquid handling speed of the robotic dispensing system (5.7 s/droplet including aspirating from bulk solution, translation of the stage and dispensing), the increase in sample number would not significantly increase the operation time of each step. As such LC-MS continues to be the bottleneck for overall throughput, as is the case with bulk analyses.

Some questions need to be additionally discussed: (1) how to control cell numbers in the droplets? Is it just randomly drawn into droplet?

We have addressed this in the manuscript as follows (Page 19): “Cell concentrations were adjusted by serially diluting them in PBS to obtain different cell numbers in nanowells. After dispensing 50 nL of cell suspension into each nanowell, we observed that the distribution of cell numbers in nanowell was stochastic, especially for low-concentration cell suspensions. Thus, the accurate cell number in each nanowell was counted using an inverted microscope and indexed to the two-dimensional spatial position of the corresponding nanowell.”

(2) The total volume of the droplets was 300 nL instead of <200 nL after added all reagents. Was part of the solutions vaporized during processing? Was it necessarily open-and-close the slide in each step?

The Reviewer raises a good question, which we have addressed in the manuscript as follows (Page 20): “During each dispensing step, the chip was opened and closed within the humidity chamber to minimize droplet evaporation. However, as the total dispensed volume in each droplet was 300 nL, and the final volume was typically <200 nL, some evaporative losses clearly occurred. Some of these water losses were observed as condensation on the contactless

cover upon cooling from the 70 °C protein extraction step, and the extended digestions at 37 °C also resulted in minor volume reductions. Such water losses have no negative effect on the performance of nanoPOTS platform, but could become limiting when further downscaling processing volumes.”

(3) Did the authors try a single cell proteome analysis using the nanoPOTS? What are the main difficulties to do this?

The prospect of extending the nanoPOTS platform to single cells is indeed exciting. We have added discussion on this topic to the manuscript as follows (Page 15): “To maximize the overall sensitivity of nanoPOTS for single cells, the total processing volume could be reduced to the low-nanoliter range to further minimize sample loss. FACS or other cell isolation techniques should be used to isolate single cells into nanowells without the minimal exogenous contamination from, e.g., secreted proteins or lysed cells. NanoLC columns with narrower bore,^{6,24} and ESI emitter technology accommodating the lower resulting flow rates⁵ could be employed to improve the detection sensitivity of the LC-MS system.”

(4) Proteome analysis for a single pancreatic islet is very attractive. Could the islet tissue samples be completely dissolved in buffer as that did same to the cultured cell samples?

The reviewer is correct that tissue samples appear to be more difficult to extract. However, in our current islet experiments, we observed minimal leftover tissue debris after extraction. Nevertheless, we will further optimize tissue extraction by exploring alternative protocols in our future studies.

(5) How about the conclusion for features of heterogeneity to different pancreatic islets?

The main purpose of our pilot single islet study is to demonstrate the feasibility of tissue analyses. The data supports the overall reproducibility and quantification as seen from the differences between T1D and control islets (Fig. 5c). We do not have a firm conclusion for features of heterogeneity and a dedicated study involving larger sample sets will be necessary. However, the addition of a comparison between healthy and type 1 diabetes patients in the revised manuscript enables a comparison between a single healthy donor and a single T1D donor.

Reviewer #4:

Kelly et al. report an exciting technological development to advance the sensitivity of bottom-up proteomics to small populations of mammalian cells. As also expressed in the manuscript, this development is important, because the transcriptome does not (necessarily) describe the phenotype of cells, cell populations, or tissue, particularly in developing system. The functionally important proteins do. However, detection of proteins in such limited amounts of samples is demands exceptional sensitivity and is subject to non-specific protein losses on vial surfaces, pipettes, etc. This work directly addresses the issue of protein losses on vial surfaces.

The new instrument that was developed in this work, termed Nanodroplet Processing in One pot for Trace Samples, or simply nanoPOTS, uses precision robotics to handle samples and reagents and microfabricated glass chips to reduce the volume of bottom-up proteomics sample processing from the typical ~100 uL to just 200 nL. Furthermore, these microchips feature designs to reduce protein adsorption, leading to sensitivity improvements. The nanoPOTS system allowed nearly all steps of the bottom-up workflow to be carried out, ranging from protein extractions, lysis, reduction, to alkylation in 1000-times less volumes. Loading of the resulting samples into closed fused silica capillaries with potential freezing of the sample was an excellent way to transfer the resulting mass-limited samples without losing proteins in this work. Coupling of the peptide digest-containing capillaries to a separation system (UPLC) followed by high-resolution mass spectrometry detection conferred excellent sensitivity. For example, ~3,000 proteins were detected and quantified over 3-log-order dynamic range with excellent quantitative reproducibility (Pearson products ~ 0.9) in just ~10 or so HeLa cells in this work. This is a record, in my view! The protein identifications were carried out with 1% FDR and 2 peptides/protein, upholding high standards set by the proteomics community. The quantified proteins and their dynamic ranges agree with those available for large-population cells, which is assuring. In addition, this study already demonstrated applicability of the tool to laser capture microdissection sampling of cells in single human pancreatic islets. These data demonstrate the usefulness of this technology not only for small cell-population analysis but also limited protein measurements.

In summary, I find the nanoPOTS system a novel development that raises important new possibilities for mass spectrometry-based proteomics. The technology is highly adaptable to existing mass spectrometry technology, and if commercialized (which I hope the researchers are already exploring), can be readily integrated into current proteomics workflows in many laboratories worldwide. The manuscript is well written using language with road audiences in mind.

I enthusiastically support publication of this work in Nature Communications. Additionally, I would like to make some suggestions and questions to the authors to help further improve this manuscript for general readers (non-experts in MS or bottom-up proteomics) – see below. I congratulate the team on this exciting technological development and look forward to seeing its success in the field.

We sincerely appreciate the Reviewer for the recognition to the contribution of our work to the proteomics field.

COMMENTS/SUGGESTIONS

1/ In the Introduction and later throughout the manuscript, it would be helpful to remind readers that – unlike in transcriptomics – bottom-up proteomics does not (yet) benefit from whole-proteome amplification. Thus, minimizing sample losses is critical to the success of proteomics, which is exactly what nanoPOTS addresses.

In the revised manuscript, we added the discussion in the introduction (Page 3) as “However, relatively large amounts of proteins from millions of cells are typically required to achieve deep

proteome coverage. Unlike genomics and transcriptomics, proteomics does not benefit from proteome amplification. Considerable efforts have thus been devoted to enhancing the overall analytical sensitivity of MS-based proteomics.”

2/ Introduction: I appreciate the authors’ reference to our single-cell proteomics work in Xenopus (Ref. 18). While, indeed, Xenopus cells are considerably large, there is an important technical detail that I would appreciate highlighted to note the significance of our study for single-cell analysis. Namely, to demonstrate the utility of our single-cell CE-ESI-MS technology to mammalian cells, Ref. 18 identified ~1,700 proteins among cells (~800 proteins/cell) by analyzing not the whole Xenopus cell’s proteome, but only a ~20 ng total protein content of the single cell, which is ~0.2% of cell’s total protein amount. Of course, this is very different than analyzing the total protein content of whole dissected single cells from Xenopus (Ref. 19).

We agree that the CE-ESI-MS technology used in Ref. 18 was demonstrated to have much higher sensitivity than LC-ESI-MS technology used in Ref 19. In the revised manuscript, we emphasized the contribution of the work by Lombard-Banek et. al. in single cell analysis (Page 4) as “Although < 0.2% of the total digest (~20 ng tryptic peptides) from single blastomeres was injected for each analysis, an identification of 500–800 protein groups in each blastomere was achieved and significant cell heterogeneity was found.¹⁸”

3/ The Slavov group recently deposited a manuscript on SCoPE-MS via bioRxiv (<https://doi.org/10.1101/102681>, title: “Mass-spectrometry of single mammalian cells quantifies proteome heterogeneity during cell differentiation”). I personally have not seen this work peer-reviewed yet, but if it has been published in a peer-reviewed format, then it could be a useful reference for this study.

We also read the manuscript by Slavov group on bioRxiv on single cell mass spectrometry. The preprint manuscript is extremely vague on the sensitivity and reproducibility and has not been published in a peer-reviewed format. Thus, it is not reasonable to discuss it in this manuscript at present.

4/ To demonstrate the sensitivity improvement in this work, it would be helpful to mention the total protein amount contained by the HeLa cells (~300 pg/cell, see ThermoFisher website) and also the amount of total peptide/protein that was successfully extracted from the cells and then analyzed by the mass spectrometer in the case of small populations of HeLa cells. Perhaps, total peptide assay could be used for this purpose. How does this protein/peptide amount compare to those analyzed in single cells in the literature?

From the website of ThermoFisher website, it indicated total protein in HeLa cell is 300 pg/cell (<https://www.thermofisher.com/us/en/home/references/ambion-tech-support/rna-tools-and-calculators/macromolecular-components-of-e.html>). However, the source of this number is not clear. A comprehensive study of protein per cell by Volpe et. al. (The FEBS Journal, 1970, 12, 195) showed the protein content in HeLa cell dynamically changed from 1570 pg/cell to 100 pg/cell during its growth cycle. The highest value was found to be 1570 pg/cell at 1 h after transferring cell to a fresh culture media at a density of 1×10^5 /mL. The protein content gradually

reduced with the increase of cell density. A relative stable value was found to be 150 pg/cell between 48 h and 72 h after changing the culture media. The value dropped to 100 pg/cell at 96 h. In our experiment, HeLa cell was split every 72 hours. Thus, the estimates protein content in single HeLa cells was 150 pg/cell.

As our response above, it will be beneficial to measure the yield of protein and peptide to directly evaluate the efficiency of sample preparation method. However, current methods to measure protein and peptide yield are not applicable to low nanogram samples. Based on cell size, HeLa cells are expected to contain similar or somewhat less protein than MCF-7 cells, which are also commonly used as a model system for small sample proteomics studies.

In the revised manuscript, we discussed the total protein content in nanoPOTS (Page 8): “Based on a quantitative study of HeLa cell growth during culture cycle,²⁸ the protein content in single HeLa cells was ~150 pg/cell. Thus, the estimated total protein contents in 10–14, 37–45 and 137–141 cells were ~1.5 ng, 6 ng, and 21 ng, respectively. The ability to identify thousands of proteins with such low-ng samples opens up new possibilities in various applications with limited sample supplies.”

5/ The islet experiments used 2 unique peptides/protein for protein identifications. Was the same setting used (2 peptides/protein) for the limited-population cell experiments?

In the revised manuscript, we added this discussion (Page 8) as “When the protein groups were constrained to contain at least two peptides, the identification numbers were 2356, 2509, and 2798 for the smallest to largest cells loadings, respectively.”

6/ Likewise, while the manuscript mentions: “Other unmentioned parameters were the default settings of the Maxquant software.” To be comprehensive and avoid potential MaxQuant-version or user-installation dependent settings in MaxQuant, please spell out the search parameters (e.g., MS1 and MS2 mass accuracy, number of allowed missed cleavages, etc.).

We thank the Reviewer for this suggestion. In the method section of the revised manuscript, we added more detail (page 22) on the Maxquant setting as “The peptide mass tolerances of the first search and main search (recalibrated) were < 20 and 4.5 ppm, respectively. The match tolerance, de novo tolerance, and deisotoping tolerance for MS/MS search were 20, 10, and 7 ppm, respectively. The minimum peptide length was 7 amino acids and maximum peptide mass was 4600 Da. The allowed missed cleavages for each peptide was 2. The second peptide search was activated to identify co-eluting and co-fragmented peptides from one MS/MS spectrum. Both peptides and proteins were filtered with a maximum false discovery rate (FDR) of 0.01. The Match Between Runs feature with a match window of 0.7 min and alignment window of 20 min was activated to increase peptide/protein identification of low-cell-number samples. LFQ calculations were performed separately in each parameter group that containing similar cell loading. Both unique and razor peptides were selected for protein quantification. Requiring MS/MS for LFQ comparisons was not activated to increase the quantifiable proteins in low-cell-number samples. Other unmentioned parameters were the default settings of the Maxquant software.”

7/ I have no doubts that the nanoPOTS system greatly minimizes protein/peptide losses during sample processing. However, one analytical question is how much role use of the new-generation mass spectrometer (Lumos) played in improving protein identifications and quantification. Based on presentations by Thermo, the Lumos improves sensitivity, charge-state identification, lower peptide redundancy, faster MSn events, etc. to boost identifications over earlier-generation mass spectrometers, which were also used in previous single-cell proteomics studies. So, to help demonstrate the improvement in relative terms, I would like to suggest a short follow-up experiment, namely to deposit the 10 or so cells into a regular Eppendorf vial and then carry out a standard bottom-up proteomics workflow on these cells using ~100 μ L total volume (for reduction, alkylation, digestion), followed by analysis of the resulting sample using the same Lumos instrument under the same experimental settings that was used in this work. This does not take much time (few days max), but it would give a relative improvement in terms of protein identifications. In turn, this would help readers appreciate the significant sensitivity improvement by nanoPOTS vs. the standard approach, which is challenged by protein losses on vial surfaces, pipettes, etc.

We thank the Reviewer for this suggestion. In the present work, the sensitivity of LC-MS was improved with smaller i.d. LC column and state of art Orbitrap Lumos MS. We agree that such improvements also contributed the peptide and protein identifications of nanoPOTS platform.

As suggested by the Reviewer, we performed the sample preparations of 10-140 HeLa cells in regular Eppendorf low-binding vial and run them using the same LC-MS setup. To provide a fair comparison, we made the following modifications for vial-based sample preparation method. (1) The total processing volume was reduced to 25 μ L, which was used in other high sensitive proteomic sample preparation method (Mol. Cell. Proteomics 14, 1672–1683 (2015)). Compared with 100 μ L volume, the smaller volume has less loss from surface absorption. It also can be directly injected into a sample loop without the concentration steps in vacuum concentrators, which could lead to potential sample loss. (2) To obtain accurate cell number in each vial, the cells were counted and lysed at high concentrations with RapiGest (5×10^5 /mL). The cell lysate was serially diluted and aliquoted into each vial for protein alkylation and digestion.

In the revised manuscript, the new result was added as Fig. 2e and 2f, and was discussed (Page 8) as “To further validate the contribution of nanoPOTS processing, we compared the performance of nanoliter versus larger volume proteomic sample processing using 10–140 HeLa cells with the exactly same LC-MS conditions. The total dispensed volume in the regular Eppendorf low-binding vials was 25 μ L, which has been used previously for highly sensitive proteomic sample preparation.⁹ The average peptide identifications ranged from 743 to 12,077, and protein identifications from 313 to 2,048 for 10, 40 and 140 cells, respectively (Fig.2e, and 2f). We note that these regular vial-based processing results were still an improvement over what has been reported previously for small-scale proteomics (Table S1), presumably due to improved LC-MS performance. However, these identifications are still significantly lower than those from nanoPOTS processing (Fig. 2c and 2d). In addition, the performance gain of nanoPOTS over vial-based preparation was much higher for the samples containing the fewest

cells. A ~25-fold increase in peptide identifications was observed for ~10-cell samples, whereas only a 2-fold increase was realized for ~140-cell samples, which further demonstrates that the nanoPOTS platform is especially beneficial for ultrasmall samples.” We feel that this additional comparison substantially strengthens the manuscript and we thank the reviewer for this suggestion.

Reviewers' comments:

Reviewer #1 (Remarks to the Author):

1) I accept the author's responses to my first question.

2) In response to my question about evaluating the accuracy of quantification, the authors wrote:

"As we show in the manuscript, the reproducibility and quantitative accuracy was evaluated by pairwise analysis of peptide and protein LFQ intensities with similar cell loadings. The correlation of protein LFQ intensities as well as the coefficients of variations (<20% for protein LFQ) were similar or even slightly better than that of our recently our recently simplified nanoproteomics platform (SNaPP) (Clair, G. et al. Sci. Rep. 2016, 6, 39223)."

This response appears to equate reproducibility and quantitative accuracy. They are not the same. Just because a method consistently provides similar estimates does not imply that the measurements are correct. As a simple example, consider the possibility that the nanoPOTS platform consistently results in lysis of about 8-10 % of the mitochondria. In such a case, the results may be highly reproducible but will not necessarily reflect the abundance of mitochondrial proteins in the cell. Since mitochondrial proteins tend to be highly abundant, they still will be identified and be represented in the GO terms analysis. There is nothing in Fig. 2 or Fig. 3 that can exclude such artifacts, e.g., differential lysis of organelles, on the quantification. Thus I still find the benchmarking of the quantification rather weak, lacking in rigour.

3) It seems that in the new revision the authors assert even more aggressively the large number of proteins identified from 10 cells, claiming over 3000 proteins from as few as 10 cells. It is essential to emphasize that the majority of these proteins are identified by using the Maxquant feature Match Between Runs (MBR). I have used MBR, and I like it. However, and crucially, peptide identification from MBR come without any statistical estimates for the confidence of identification. Thus the number of identified proteins by MBR is hardly a solid number on which to stake a claim of coverage depth. It lacks formal statistical estimate of false discovery rate.

4) The authors have improved their analysis of the data from the pancreatic islets (Fig. 5c), but it still seems unsystematic and unclear how they quantify heterogeneity within a pancreatic islet. I assume the claim of measuring heterogeneity is based on the protein B2M, last column in Fig. 5c. If each measurement averages the protein levels of 90 - 454 cells, why do we expect to see heterogeneity in such large groups of cells. This seems at odds with previous theoretical and empirical estimates, e.g., Janes KA, et al. Identifying single-cell molecular programs by stochastic profiling. Nat Methods 7:311–317 (2010)

Reviewer #2 (Remarks to the Author):

I requested the original files and have searched these using the software and settings used by the authors. Indeed using the raw data it is possible to identify ~1,500 proteins per a sample of ~10 cells. However, of these only about 1,000 proteins were identified with at least two peptides. In addition, only 10% of the identified proteins are GO-annotated as 'integral to membrane'. This indicates substantial bias because membrane proteins account to about 30% of the proteome. Using LC-MS/MS settings provided by the authors I haven't been able to achieve the sensitivity reported in the manuscript. Using QExactive HF instrument, at least ten times more protein digest was necessary to attain results comparable with the reported ones. Might be the instrument used by authors performed much better than instruments I used, but this also could be the case in majority of laboratories trying to use the 'Nanodroplet' technology. In conclusion, I do not believe that such data covering 1000-1500 proteins from 10 cells are useful for studying proteomes or identification of biomarkers. In my opinion, the 'Nanodroplet' technology is not a breakthrough in proteomics and in a 10 of 30 cell format allows identification of mainly highly abundant, housekeeping proteins. In addition, mass spectrometer sensitivity reported in the paper will be difficult to reproduce.

Reviewer #3 (Remarks to the Author):

The revised manuscript is clearly demonstrated for all of the questions. It is recommended to be accepted for publication.

Reviewer #4 (Remarks to the Author):

In this revision, the authors have address my main concerns related to this manuscript. I also thank the Authors for carrying out the vial-based digestion study as a control, which now serves as validation to the sensitivity gain that is obtained by NanoPOTS due to minimized surface adsorptive losses of proteins. Additionally, this revised version of the manuscript has addressed with additional tables and evaluations the excellent comments that were provided by the other Reviewers earlier. I am enthusiastic of this work, because of its innovative approach to advance bottom-up proteomics sensitivity for trace amounts of samples. It is exciting to think about the potentials that NanoPOTS can raise in proteomics in many laboratories, particularly if the platform is commercialized to facilitate broad adaptation.

Point-by-Point Response to the Reviewer Comments

Reviewer comments in italics, responses in blue.

Reviewer #1 (Remarks to the Author):

1) I accept the author's responses to my first question.

These first questions from the previous critique related to characterization of the throughput, robustness and potential for scale-up of the nanoPOTS platform, as well as the substantial increase in proteome coverage that nanoPOTS enables relative to past approaches. We are glad that the reviewer now accepts these critical aspects of our work.

2) In response to my question about evaluating the accuracy of quantification, the authors wrote:

"As we show in the manuscript, the reproducibility and quantitative accuracy was evaluated by pairwise analysis of peptide and protein LFQ intensities with similar cell loadings. The correlation of protein LFQ intensities as well as the coefficients of variations (<20% for protein LFQ) were similar or even slightly better than that of our recently our recently simplified nanoproteomics platform (SNaPP) (Clair, G. et al. Sci. Rep. 2016, 6, 39223)."

This response appears to equate reproducibility and quantitative accuracy. They are not the same. Just because a method consistently provides similar estimates does not imply that the measurements are correct. As a simple example, consider the possibility that the nanoPOTS platform consistently results in lysis of about 8-10 % of the mitochondria. In such a case, the results may be highly reproducible but will not necessarily reflect the abundance of mitochondrial proteins in the cell. Since mitochondrial proteins tend to be highly abundant, they still will be identified and be represented in the GO terms analysis. There is nothing in Fig. 2 or Fig. 3 that can exclude such artifacts, e.g., differential lysis of organelles, on the quantification. Thus I still find the benchmarking of the quantification rather weak, lacking in rigour.

We agree with the reviewer that our primary focus has been on relative quantification using label-free quantification (LFQ) with the open-source MaxQuant data analysis platform (Cox and Mann, 2008, *Nature Biotech*, DOI: 10.1038/nbt.1511; Cox et al., 2016, *Nature Protocols*, DOI: 10.1038/nprot.2016.136). We are glad that the reviewer appears to accept that this aspect is well demonstrated. The vast majority of current proteomics studies rely on relative quantification approaches such as LFQ. We demonstrated the reproducibility of nanoPOTS for LFQ and its ability to identify differentially expressed proteins in clinical samples as follows:

- Pairwise correlation of LFQ intensities for cultured cells (Figure 4)
- Coefficients of variance for LFQ intensities for cultured cells (Figure 4)
- Pairwise correlation of LFQ intensities for single islet sections (Figure 5)
- Comparison of protein abundances in single islets between a non-diabetic donor and a type 1 diabetes donor (Figure 5)
- Additional peptide-level and protein-level pairwise correlations in Figures S8 and S9, respectively
- Additional LFQ quantitation reproducibility in Fig. S10

- A volcano plot identifying 169 proteins that are differentially expressed in non-diabetic and T1D donors based on LFQ in Fig. S13

However, the reviewer raised a good concern regarding the potential of “*absolute*” quantification, about which we make no claims. While the aspect of absolute quantification is of interest, we recognize that most current proteomics approaches suffer from different degrees of biases such as protein extraction as raised by the reviewer, digestion, ionization suppression, and others, which limits their application to relative abundance changes. Therefore, it is not our intent or focus to emphasize the aspect of absolute quantification using the nanoPOTS platform at its current stage. Nevertheless, the distribution of proteins across different subcellular organelles is very comparable between nanoPOTS and a bulk protocol (Figure S12), suggesting there is minimal extraction bias if any for different organelles.

To avoid any misunderstanding, we clarified the emphasis on relative quantification on page 11: “To assess whether the nanoPOTS platform can provide comparative quantitative proteome profiling, its reproducibility using MS1 intensity measurements was assessed at both the peptide and protein levels using label-free quantification.”

3) It seems that in the new revision the authors assert even more aggressively the large number of proteins identified from 10 cells, claiming over 3000 proteins from as few as 10 cells. It is essential to emphasize that the majority of these proteins are identified by using the Maxquant feature Match Between Runs (MBR). I have used MBR, and I like it. However, and crucially, peptide identification from MBR come without any statistical estimates for the confidence of identification. Thus the number of identified proteins by MBR is hardly a solid number on which to stake a claim of coverage depth. It lacks formal statistical estimate of false discovery rate.

We would like to clarify that our data regarding proteome coverage has not changed since our initial submission, and in each case in our manuscript, we differentiate between protein groups identified by MS/MS alone vs. MS/MS plus MBR, so the identifications using both approaches are easily distinguished.

The developers of MaxQuant assert that the MBR feature is indeed FDR-controlled. As stated in the developers’ web description: “An **FDR-controlled algorithm called matching between runs** is incorporated, which enables the MS/MS free identification of MS features in the complete data set for each single measurement. This leads to an increase in the number of quantified proteins per sample”

(<http://www.biochem.mpg.de/5111795/maxquant>, emphasis added). Mann and coworkers also provide details for their FDR evaluation of MBR analyses across 3 different cell lines (Molecular & Cellular Proteomics, 11(3), M111-014050), which was found to be considerably lower than 1% (page 4, right column).

In addition, Prof. Mann, a leader in the field of proteomics, has strongly recommended the use of MBR for small-sample proteomics [2, 3] and in-depth proteomics [1, 4, 5], and has demonstrated its success for enabling new biological discoveries [1-5]. For example, in his recent paper on single (large) muscle cell proteomics published in *Cell Reports* [2], at least 40% of quantifiable proteins were identified using MBR (Figure S2B).

To further evaluate our MBR results, we employed a quality control software “PTXQC” (*Journal of Proteome Research*, 15(3), 777-787). PTXQC introduced two metrics to score MaxQuant’s MBR functionality including MBR alignment and ID-transfer. We discussed this in the revised manuscript on page 8: “We further employed an open-source quality control software to evaluate the quality of the MBR identifications.²⁶ For all datasets used in this study, both MBR alignment and ID-transfer metrics (Fig. S6) indicate high confidence of the MBR-transferred identifications.”

References for this comment:

- [1]. Sharma, K., Schmitt, S., Bergner, C. G., Tyanova, S., Kannaiyan, N., Manrique-Hoyos, N., ... & Rossner, M. J. (2015). **Nature neuroscience**, 18(12), 1819-1831.
- [2]. Murgia, M., Toniolo, L., Nagaraj, N., Ciciliot, S., Vindigni, V., Schiaffino, S., ... & Mann, M. (2017). **Cell Reports**, 19(11), 2396-2409.
- [3]. Murgia, M., Nagaraj, N., Deshmukh, A. S., Zeiler, M., Cancellara, P., Moretti, I., ... & Mann, M. (2015). **EMBO reports**, 16(3), 387-395.
- [4]. Geiger, T., Wehner, A., Schaab, C., Cox, J., & Mann, M. (2012). **Molecular & Cellular Proteomics**, 11(3), M111-014050.
- [5]. Kulak, N. A., Geyer, P. E., & Mann, M. (2017). **Molecular & Cellular Proteomics**, 16(4), 694-705.

4) The authors have improved their analysis of the data from the pancreatic islets (Fig. 5c), but it still seems unsystematic and unclear how they quantify heterogeneity within a pancreatic islet. I assume the claim of measuring heterogeneity is based on the protein B2M, last column in Fig. 5c. If each measurement averages the protein levels of 90 - 454 cells, why do we expect to see heterogeneity in such large groups of cells. This seems at odds with previous theoretical and empirical estimates, e.g., Janes KA, et al. Identifying single-cell molecular programs by stochastic profiling. Nat Methods 7:311–317 (2010).

This comment appears to reflect a misunderstanding. We are not investigating heterogeneity within a single human islet such as those of single cell technologies and make no claims to that effect. The 9 islet sections from a healthy donor, and the 9 islet sections from a type 1 diabetes donor are all from different islets. As such, we are not comparing heterogeneity within an islet but rather potential differences between T1D and controls through single islet measurements. Again, we emphasize that the pilot study is to demonstrate the feasibility of the nanoPOTS platform for tissue-based biomedical research and to quantify differences between two states, which the reviewer does not dispute. While islet heterogeneity is not the focus of this pilot study, such heterogeneity is well known in type 1 diabetes (see Ref 31 in the manuscript).

Reviewer #2 (Remarks to the Author):

I requested the original files and have searched these using the software and settings used by the authors. Indeed using the raw data it is possible to identify ~1,500 proteins per a sample of ~10 cells. However, of these only about 1,000 proteins were identified with at least two peptides.

We thank the reviewer for independently processing our raw data and validating that indeed ~1,500 protein groups can be identified by MS/MS for our smallest sample loadings containing only ~10 cells. We included an FDR of 1% and accepted protein identification based on a single peptide. We should note that all past “small-sample” studies used for comparison to the nanoPOTS platform (References 9, 12-14, 17 in manuscript) also accepted single peptide identifications. This single peptide per protein criteria has been routinely applied in the field [1-8], including by Professor Norman Dovichi, a leading expert in ultrasensitive proteomic analyses. Therefore, we believe our practice is in line with established protocols.

We should also note that, as described above, there is an increasing trend to incorporate the MBR algorithm for high sensitivity proteomic studies, which is well-demonstrated by, e.g., the Matthias Mann group and Norman

Dovichi group [2, 8]. In our work, with MBR, the average protein identifications is 3,092 for samples comprising ~10 cells, and 2,674 protein groups were quantifiable across three biological replicates.

However, it is also of some value to report that the majority of MS/MS-identified proteins were also identified using more stringent criteria based on a minimum of two-peptides. We have added the following sentence on p. 7: “The average number of protein groups were 965 to 2,167 when at least two peptides were required for identification.”

References for this comment:

1. Sun, L., G. Zhu, Y. Li, P. Yang, and N.J. Dovichi, Coupling methanol denaturation, immobilized trypsin digestion, and accurate mass and time tagging for liquid-chromatography-based shotgun proteomics of low nanogram amounts of RAW 264.7 cell lysate. *Anal Chem*, 2012. 84(20): p. 8715-8721.
2. Sun, L., G. Zhu, Y. Zhao, X. Yan, S. Mou, and N.J. Dovichi, Ultrasensitive and fast bottom-up analysis of femtogram amounts of complex proteome digests. *Angewandte Chemie*, 2013. 52(51): p. 13661-13664.
3. Yan, X., D.C. Essaka, L. Sun, G. Zhu, and N.J. Dovichi, Bottom-up proteome analysis of *E. coli* using capillary zone electrophoresis-tandem mass spectrometry with an electrokinetic sheath-flow electrospray interface. *Proteomics*, 2013. 13(17): p. 2546-2551.
4. Sun, L., G. Zhu, S. Mou, Y. Zhao, M.M. Champion, and N.J. Dovichi, Capillary zone electrophoresis-electrospray ionization-tandem mass spectrometry for quantitative parallel reaction monitoring of peptide abundance and single-shot proteomic analysis of a human cell line. *Journal of chromatography. A*, 2014. 1359: p. 303-308.
5. Ludwig, K.R., L.L. Sun, G.J. Zhu, N.J. Dovichi, and A.B. Hummon, Over 2300 Phosphorylated Peptide Identifications with Single-Shot Capillary Zone Electrophoresis-Tandem Mass Spectrometry in a 100 min Separation. *Anal. Chem.*, 2015. 87(19): p. 9532-9537.
6. Sun, L.L., G.J. Zhu, Z.B. Zhang, S. Mou, and N.J. Dovichi, Third-Generation Electrokinetically Pumped Sheath-Flow Nanospray Interface with Improved Stability and Sensitivity for Automated Capillary Zone Electrophoresis-Mass Spectrometry Analysis of Complex Proteome Digests. *Journal of Proteome Research*, 2015. 14(5): p. 2312-2321.
7. Zhang, Z.B., L.L. Sun, G.J. Zhu, O.F. Cox, P.W. Huber, and N.J. Dovichi, Nearly 1000 Protein Identifications from 50 ng of *Xenopus laevis* Zygote Homogenate Using Online Sample Preparation on a Strong Cation Exchange Monolith Based Microreactor Coupled with Capillary Zone Electrophoresis. *Anal. Chem.*, 2016. 88(1): p. 877-882.
8. Sun, L.L., K.M. Dubiak, E.H. Peuchen, Z.B. Zhang, G.J. Zhu, P.W. Huber, and N.J. Dovichi, Single Cell Proteomics Using Frog (*Xenopus laevis*) Blastomeres Isolated from Early Stage Embryos, Which Form a Geometric Progression in Protein Content. *Anal. Chem.*, 2016. 88(13): p. 6653-6657.

In addition, only 10% of the identified proteins are GO-annotated as ‘integral to membrane’. This indicates substantial bias because membrane proteins account to about 30% of the proteome.

We thank the reviewer for raising this point. However, any bias with respect to underrepresented membrane proteins is clearly not due to any special issue with our processing platform as demonstrated by the nearly identical representation using the completely distinct SNaPP preparation and analysis method, which utilized a 8M Urea-based full extraction protocol (See Figure S12), but rather the result of membrane proteins typically being present in lower copy numbers than other proteins, their relatively poor solubility, and a paucity of available trypsin cleavage sites. With further improvement of the overall sensitivity and proteome coverage of

our technology, we anticipate the coverage of membrane proteins will enhance. Alternatively, to increase the coverage of membrane proteins, special protein extraction, enrichment, and fractionation approaches are required. A number of research and review papers from leading proteomic experts have focused on the challenges of membrane proteomics (Nature biotechnology, 21(3), 262-267, Journal of proteome research, 15(4), 1243-1252, Proteomics, 8(19), 3924-3932, Journal of proteomics, 15(3), 8-20, Proteomics, 13(3-4), 404-423).

Using LC-MS/MS settings provided by the authors I haven't been able to achieve the sensitivity reported in the manuscript. Using QExactive HF instrument, at least ten times more protein digest was necessary to attain results comparable with the reported ones. Might be the instrument used by authors performed much better than instruments I used, but this also could be the case in majority of laboratories trying to use the 'Nanodroplet' technology.

We are grateful for the reviewer's interest for replicating our results. The reviewer found that ~10 times more protein digest was required to achieve coverage similar to what we achieved for our smallest samples (~10 cells or ~2 ng protein). While we agree sometimes the sensitivity level maybe slightly different between LC-MS platforms, we should note that our LC-MS platforms are completely based on commercially available products and we do not claim any unique aspects of the LC-MS platform itself. Furthermore, the raw analytical sensitivity and proteome coverage that we report is in line with our own past results and those reported by others. For example, we recently analyzed human and microbial lysates in the range of 0.5 ng to 50 ng and achieved robust and high proteome coverage (DOI: 10.1016/j.ijms.2017.08.016). Similarly, Kocher et al. identified >2,000 proteins from 10 ng of HeLa lysate using MS/MS alone and a previous generation MS instrument (DOI 10.1002/pmic.201300418). And as we mention in our manuscript, Li et al. achieved in-depth proteome coverage from nanogram sample loadings (~2000 protein groups using 100 MCF-7 lysate in Figure 3) (DOI: 10.1074/mcp.M114.045724).

We believe the LC-MS platforms can be fully benchmarked between different labs for its performance as demonstrating in recent CPTAC (Clinical Proteomics for Tumor Analysis Consortium). As shown in the following figure the achieved proteome coverage or sensitivity is reproducible between the Broad Institute, Johns Hopkins

and PNNL platforms (unpublished results).

Other points for consideration:

- The reviewer used a previous generation Q Exactive HF instrument for comparison. We do not know the exact difference in sensitivity between this instrument and the Lumos for small samples. As that instrument lacks an electrodynamic ion funnel that was incorporated into the Lumos instrument that we

used, it would be expected to have somewhat reduced ion transmission efficiency and sensitivity compared to the Lumos, which could account for a difference in coverage.

- The reviewer gives no indication whether the same custom-made 30- μm -i.d. packed liquid chromatography column with an integrated electrospray emitter was used for his or her tests. The miniaturized LC was crucial to achieving the required sensitivity and proteome coverage that we reported.
- There are many subtle ways in which a complex LC/MS system may be operated under less than optimal conditions. As such, achieving similar coverage for sample sizes differing by just a factor of 10 across laboratories and with different instrumentation should serve as validation of our results rather than calling them into question.

In conclusion, I do not believe that such data covering 1000-1500 proteins from 10 cells are useful for studying proteomes or identification of biomarkers. In my opinion, the 'Nanodroplet' technology is not a breakthrough in proteomics and in a 10 of 30 cell format allows identification of mainly highly abundant, housekeeping proteins. In addition, mass spectrometer sensitivity reported in the paper will be difficult to reproduce.

We appreciate the reviewer's efforts in providing a detailed assessment of our data and in replicating our results. However, we do not quite agree with the logic of this conclusion. In the preceding comment, the reviewer seems to assert that our sensitivity and coverage are so high that no one can reproduce them, and now the claim appears to be that our sensitivity is still too low to be useful for proteomic studies.

On the contrary, we would like to point out the following:

- 1) The reviewer's effort on analyzing our raw data validated our coverage for 10 cells independently.
- 2) The reviewer would agree that we utilized a commercial LC-MS platform, which should be relatively reproducible across different labs. Unfortunately, the reviewer did not use the same platform for his or her assessment.
- 3) NanoPOTS was clearly demonstrated as a breakthrough technology for small sample processing as illustrated in Fig. 2 comparing vial-based processing vs. nanoPOTS
- 4) The assertion that our platform only identified highly abundant, housekeeping proteins was refuted by our own pilot study. Our analysis of laser-dissected single islet sections from type 1 diabetes (T1D) and non-diabetic donors revealed 169 differentially expressed proteins, many of which are highly relevant to pancreatic islet function and type 1 diabetes such as MHC class I antigens (Figure 5). Of course, further improvements in protein coverage are always desirable and are the focus of future optimization and development efforts.
- 5) Finally, the coverage the reviewer states in this comment (1000-1500 proteins) seems to be ignoring the proteins identified by MBR, which produced a substantial increase in coverage to >3000 protein groups for our smallest samples. As we mention above, it has been widely recognized for more than a decade that identifying peptides based on their intact masses and normalized elution times can help to overcome the clear limitations associated with relying on MS/MS alone in terms of sequencing throughput and sensitivity. For this reason, our group developed the Accurate Mass and Time (AMT) tag approach, and Matthias Mann and coworkers developed the MBR approach. Both approaches have been widely used and validated for substantially enhancing proteome coverage.

Reviewer #3 (Remarks to the Author):

The revised manuscript is clearly demonstrated for all of the questions. It is recommended to be accepted for publication.

Reviewer #4 (Remarks to the Author):

In this revision, the authors have address my main concerns related to this manuscript. I also thank the Authors for carrying out the vial-based digestion study as a control, which now serves as validation to the sensitivity gain that is obtained by NanoPOTS due to minimized surface adsorptive losses of proteins. Additionally, this revised version of the manuscript has addressed with additional tables and evaluations the excellent comments that were provided by the other Reviewers earlier. I am enthusiastic of this work, because of its innovative approach to advance bottom-up proteomics sensitivity for trace amounts of samples. It is exciting to think about the potentials that NanoPOTS can raise in proteomics in many laboratories, particularly if the platform is commercialized to facilitate broad adaptation.

We thank both Reviewers 3 and 4 for their strong support of our work.

Reviewers' comments:

Reviewer #1 (Remarks to the Author):

The confusion over relative versus absolute quantification stems from misunderstanding of how "coefficients of variance (CVs)" are defined. Coefficients of variation (I have not seen before "coefficients of variance" that the authors use) are defined as standard deviation / mean. However, the authors did not describe how they compute them. I assume they did so for fold changes of the same protein estimated from different peptides. If so, the CVs of all proteins IDed with a single peptide are zero by definition, which will contribute to substantial underestimation of their CVs and thus error. Furthermore, in their "validation" all protein fold changes are expected to be 1, and thus CVs in this study, however underestimated, at best can support that the authors can see when proteins are not changing in abundance. Spiked in controls are required to show that they can identify proteins changing their abundances in expected ways.

As I explained before, I find the correlations between LFQ intensities useless in evaluating relative quantification. See some of the statistical problems with this here:
<https://www.nature.com/nature/journal/v547/n7664/full/nature23293.html>

I suggest to the authors to examine the posterior error probabilities and the false discovery rate (q values) estimated by the Match Between Runs (MBR) algorithms returned by Maxquant in the evidence file.

Reviewer #2 (Remarks to the Author):

Response to the author comments:

In my recent report I expressed my doubts on the value of the nanoPOTS technology. In the responses authors very thoroughly replayed to my criticism. I agree that it is difficult to compare performances of LC-MS/MS between laboratories but I have still to major concerns.

1. It is not a rule that membrane proteins occur at low copy numbers compared to cytosolic protein. This may be true for plasma membrane but not in general considering abundant organelles such as reticulum of Golgi. In the response authors suggest that further "improvements of the overall sensitivity and proteome coverage of our technology we anticipate the coverage of membrane proteins will enhance. " At the moment this can be considered solely as a promise.

2. I agree that using MBR the number of identified proteins per sample can be significantly increase. Using MBR more samples, means more cells, are required. Thus, for the sake of clarity

it should be stated that using 10 cells nanoPOTS platform allows identification of 1000-1500 proteins, but not 3000.

Reviewer #5 (Remarks to the Author):

The authors describe a novel and optimized method for the sample processing and analysis of proteins from very low amounts of cells, nanoPOTS. The protocol is an automated and miniaturized version of the RapiGest one-pot protocol. They report between ~1500 and ~3000 protein identifications for triplicate measurements for groups with ~12, ~40 or ~140 cells. To obtain these numbers, they applied match-between-runs (MBR) label-free analysis using MaxQuant. To validate their results, the authors compare several measurements, both quantitative and qualitative. The results of all these comparisons appear to have little bias introduced by the low amounts of protein measured, other than an expected overrepresentation of more abundant proteins.

Even though I believe the quantified numbers may be inflated because of the match between runs method and imputation, it is clear that the method works to extract and identify proteins from very low numbers of cells, and that these results are reproducible. I do have some issues that should be addressed before publication:

Major concerns

In the comparison of islet cells between T1 diabetic donors and healthy controls, the authors report a significant the number of 169 significant proteins (line 290 and Figure S13). There is a discrepancy: the method (line 557) describes a FDR threshold of 0.01, and fold change threshold of 2x, while Figure S13 appear to contain an uncorrected p-value cutoff of 0.05 combined with a fold change cutoff of 4x. Please correct this discrepancy, since now it is unclear where the number of significant proteins comes from.

Apart from that, an unknown number of quantified proteins is assigned a value using imputation. Since match-between-runs is already used to increase the number of identifications and quantified values, another opportunity to obtain false positive results is introduced. Therefore, it is important to show how many, and which proteins are significant because of this imputation. It would be commendable to re-analyze the data without the imputed values and report these numbers alongside the currently reported numbers.

Minor concerns

Figure 2b: because of the scaling it has become much less easy to compare the elution patterns

between the methods. It appears the pattern becomes more complex, but with an increased intensity. Preferably scale the plots in such a way that the patterns are clearer.

Figure 5c. This plot may not be readable for color-blind readers. Moreover, the choice of the color green for the diseased condition and red for the control could lead to mis-interpretation of the figure.

In Figure S6 the text is completely unreadable.

Point-by-Point Response to the Reviewer Comments

Reviewer comments in italics, responses in blue.

Reviewer #1 (Remarks to the Author):

The confusion over relative versus absolute quantification stems from misunderstanding of how "coefficients of variance (CVs)" are defined. Coefficients of variation (I have not seen before "coefficients of variance" that the authors use) are defined as standard deviation / mean. However, the authors did not describe how they compute them. I assume they did so for fold changes of the same protein estimated from different peptides. If so, the CVs of all proteins IDed with a single peptide are zero by definition, which will contribute to substantial underestimation of their CVs and thus error. Furthermore, in their "validation" all protein fold changes are expected to be 1, and thus CVs in this study, however underestimated, at best can support that the authors can see when proteins are not changing in abundance. Spiked in controls are required to show that they can identify proteins changing their abundances in expected ways.

We are in agreement with the Reviewer on the definition of coefficients of variation (CVs), and we defined CVs in the in the 'Data analysis' section of our manuscript as follows: "Coefficients of variation (CVs) were defined as the standard deviation of normalized intensities divided by the mean intensity across the processing replicates of the same loading". We have corrected the two instances where 'variance' was used in place of 'variation'.

To further clarify, we did not compute the CVs as fold changes of the same protein estimated from different peptides as the reviewer assumed. CVs are calculated using the LFQ abundances at the protein level between processing replicates (doi: 10.1074/mcp.M113.031591), and no underestimation is anticipated. Furthermore, our observed CVs are at similar levels compared to bulk analyses.

In short, since our coefficients of variation are defined as standard deviation/mean from processing replicates, the CVs provide a good demonstration of the technical reproducibility of the overall workflow, hence its applicability for relative quantification between different biological conditions. Numerous papers show that label-free proteomics is quantitative under properly controlled experimental conditions as done here, and reproducing spike-in experiments does not seem to be necessary (doi: 10.1074/mcp.M113.031591, , doi: 10.1002/prca.201400202, , doi: 10.1074/mcp.R112.025163, doi: 10.1016/j.bbapap.2013.04.001, doi: 10.1074/jbc.R110.199703).

As I explained before, I find the correlations between LFQ intensities useless in evaluating relative quantification. See some of the statistical problems with this here:
<https://www.nature.com/nature/journal/v547/n7664/full/nature23293.html>

Both correlations and coefficients of variation were used to evaluate the technical reproducibility of the overall workflow. Such evaluation has been widely used in numerous of label free protein quantification studies [1-4]. We understand the Reviewer's concern about the correlation plots of the LFQ intensities, and we acknowledge such plots are not a perfect evaluation of reproducibility since the Pearson correlation coefficients can be highly impacted by the highest intensity data points.

For this reason, we also plotted the CV distributions (violin plots) along with the correlation plots as Figure 4d. Furthermore, we have included a Supplemental Figure to illustrate the CV versus intensity (Figure S12) in which ~20% mean CVs were observed even for low-intensity data.

Therefore, we believe we have carefully addressed the reproducibility issue for our overall workflow, and the ability to detect significant abundance differences was demonstrated in the single islet study between T1D and control samples (Figure 5).

References for this comment:

- [1]. Sharma, K., Schmitt, S., Bergner, C. G., Tyanova, S., Kannaiyan, N., Manrique-Hoyos, N., ... & Rossner, M. J. (2015). **Nature neuroscience**, 18(12), 1819-1831.
- [2]. Lombard-Banek, C.; Reddy, S.; Moody, S. A.; Nemes, P. **Mol. Cell. Proteomics** 2016, 15 (8), 2756–2768.
- [3]. Clair, G.; Piehowski, P. D.; Nicola, T.; Kitzmiller, J. A.; Huang, E. L.; Zink, E. M.; Sontag, R. L.; Orton, D. J.; Moore, R. J.; Carson, J. P.; Smith, R. D.; Whitsett, J. A.; Corley, R. A.; Ambalavanan, N.; Ansong, C. **Sci. Rep.** 2016, 6, 39223.
- [4]. Murgia, M., Toniolo, L., Nagaraj, N., Ciciliot, S., Vindigni, V., Schiaffino, S., ... & Mann, M. (2017). **Cell Reports**, 19(11), 2396-2409.
- [5] Ståhl, P. L.; Salmén, F.; Vickovic, S.; Lundmark, A.; Navarro, J. F.; Magnusson, J.; Giacomello, S.; Asp, M.; Westholm, J. O.; Huss, M.; Mollbrink, A.; Linnarsson, S.; Codeluppi, S.; Borg, Å.; Pontén, F.; Costea, P. I.; Sahlén, P.; Mulder, J.; Bergmann, O.; Lundeberg, J.; Frisén, J. **Science** 2016, 353 (6294), 78–82.

I suggest to the authors to examine the posterior error probabilities and the false discovery rate (q values) estimated by the Match Between Runs (MBR) algorithms returned by Maxquant in the evidence file.

We acknowledge that the false discovery rate is a concern for MBR. In our last submission, we implemented quality control software “PTXQC” (*Journal of Proteome Research*, 15(3), 777-787) to evaluate the MBR identifications. PTXQC uses two metrics to score MaxQuant’s MBR functionality including MBR alignment and ID-transfer. For all datasets used in this study, both MBR alignment and ID-transfer metrics (Fig. S7) indicate high confidence of the MBR-transferred identifications.

Furthermore, we have checked the MaxQuant evidence file. While we have been unable to find the posterior error probabilities and the false discovery rate (q values) in the evidence file, we decided to plot the mass error distribution and retention time distribution histograms for MBR data points as well as those data points with MS/MS IDs. The comparable distributions of mass errors between MBR and MS/MS data points as well as the tight distribution of retention time differences are indicative of high-confidence matching. Also, as demonstrated in our previous work (DOI: 10.1021/pr800467r), the background random match level can be used to estimate the false discovery rate (see figures below). Using this approach, we estimate the FDR is ~2%. This low FDR is partly due to the high mass accuracy and retention time accuracy of the advanced LC-MS system.

Finally, we did seek input from Prof. Jürgen Cox, who is the developer of MaxQuant. His reply was as follows:

*“We have an **internal method** for estimating false positives based on adding decoy peaks with shifted masses. It looks like FDRs are typically around 2-4%, but that depends on the sample. Furthermore, **the MBR FDR is a MaxQuant feature that we are working on, but it is not ready to use, yet.**” (emphases added)*

Therefore, our estimated FDR is largely in line with Dr. Cox’s assessment and confirms that q values are not currently provided by the evidence file.

Reviewer #2 (Remarks to the Author):

In my recent report I expressed my doubts on the value of the nanoPOTS technology. In the responses authors very thoroughly replayed to my criticism. I agree that it is difficult to compare performances of LC-MS/MS between laboratories but I have still to major concerns.

We are happy to see that this Reviewer is recognizing the value of the nanoPOTS platform.

1. It is not a rule that membrane proteins occur at low copy numbers compared to cytosolic protein. This may be true for plasma membrane but not in general considering abundant organelles such as reticulum of Golgi. In the response authors suggest that further “improvements of the overall sensitivity and proteome coverage of our technology we anticipate the coverage of membrane proteins will enhance. “ At the moment this can be considered solely as a promise.

We perfectly agree with the Reviewer comment. As we responded previously, any underrepresentation of membrane proteins may arise from various factors that are common across proteomics workflows. These factors include low copy numbers, poor solubility, low extraction efficiency and a paucity of trypsin cleavage sites. Any of these problems can lead to low amounts of peptides from membrane proteins for MS analysis. However, we are quite confident our processing platform itself does not introduce particular bias towards membrane proteins. We observed nearly identical representation using the completely distinct SNaPP preparation and analysis method, which utilized an 8M urea-based full extraction protocol (See Figure S15).

2. I agree that using MBR the number of identified proteins per sample can be significantly increase. Using MBR more samples, means more cells, are required. Thus, for the sake of clarity it should be stated that using 10 cells nanoPOTS platform allows identification of 1000-1500 proteins, but not 3000.

Following the Reviewer's suggestion, we have revised the abstract as follows to provide further clarification: "When combined with ultrasensitive LC-MS, nanoPOTS allows identification of ~1,500 and ~3,000 proteins from ~10 and ~140 cells, respectively. By incorporating the Match Between Runs algorithm of MaxQuant, >3000 proteins were consistently identified from as few as 10 cells."

Reviewer #5 (Remarks to the Author):

The authors describe a novel and optimized method for the sample processing and analysis of proteins from very low amounts of cells, nanoPOTS. The protocol is an automated and miniaturized version of the RapiGest one-pot protocol. They report between ~1500 and ~3000 protein identifications for triplicate measurements for groups with ~12, ~40 or ~140 cells. To obtain these numbers, they applied match-between-runs (MBR) label-free analysis using MaxQuant. To validate their results, the authors compare several measurements, both quantitative and qualitative. The results of all these comparisons appear to have little bias introduced by the low amounts of protein measured, other than an expected overrepresentation of more abundant proteins.

Even though I believe the quantified numbers may be inflated because of the match between runs method and imputation, it is clear that the method works to extract and identify proteins from very low numbers of cells, and that these results are reproducible. I do have some issues that should be addressed before publication:

We sincerely appreciate the Reviewer for the recognition to the contribution of our work.

Major concerns

In the comparison of islet cells between T1 diabetic donors and healthy controls, the authors report a significant the number of 169 significant proteins (line 290 and Figure S13). There is a discrepancy: the method (line 557) describes a FDR threshold of 0.01, and fold change threshold of 2x, while Figure S13 appear to contain an uncorrected p-value cutoff of 0.05 combined with a fold change cutoff of 4x. Please correct this discrepancy, since now it is unclear where the number of significant proteins comes from.

We appreciate the Reviewer for pointing out this discrepancy. We have corrected this to make the criteria consistent between the text and supplemental figures. Briefly, to identify significant changes of single islets between the T1D donor and a healthy control, we employed a Student's t-test with the Permutation-based FDR control approach embedded in Perseus platform (DOI: 10.1038/nmeth.3901). The Permutation-based FDR control was chosen because it is a preferred method for situations in which few biological replicates ($n < 10$) are obtained, which is common in proteomic studies due to the high cost per analysis. The significant proteins were defined by requiring uncorrected p-value < 0.01 , q-value (FDR) < 0.05 , and $\log_2(\text{fold change}) > 1$. The FDR estimation was based on the SAM method (Significant Analysis of Microarrays) (DOI: 10.1073/pnas.091062498).

We have added more details on the approach used for statistical analysis as follows: “A Student’s t-test with permutation-based FDR control approach was used for statistical analysis. A p-value <0.01, q-value <0.05 and a fold change >2 were used to identify proteins with significant abundance changes between T1D and control.”

Apart from that, an unknown number of quantified proteins is assigned a value using imputation. Since match-between-runs is already used to increase the number of identifications and quantified values, another opportunity to obtain false positive results is introduced. Therefore, it is important to show how many, and which proteins are significant because of this imputation. It would be commendable to re-analyze the data without the imputed values and report these numbers alongside the currently reported numbers.

We appreciate the Reviewer’s comment and indeed we agree this is a potential concern. Following the Reviewer suggestion, we have further analyzed the data without imputed values using an independent G-test with multiple testing correction using the Benjamini-Hochberd method (DOI: 10.1021/pr0600273). Among the 333 significant proteins, we have 289 proteins with at least one missing data point across the 18 single islet samples. Among the 289, with G-test we have 137 proteins with unadjusted p-value <0.05 and 64 proteins with adjusted p-value <0.05. It was anticipated that without imputation the number of significant proteins would be much lower. However, the directions of changes for all 137 proteins are concordant between data with imputation (t-test) and without imputation (G-test). The data suggest that the results from imputation are robust. We have included this additional information in our revised manuscript text (Figure S16 notes).

Minor concerns

Figure 2b: because of the scaling it has become much less easy to compare the elution patterns between the methods. It appears the pattern becomes more complex, but with an increased intensity. Preferably scale the plots in such a way that the patterns are clearer.

Following the suggestion of the Reviewer, in the revised manuscript, we added Fig. S4 with full-scaled Y axis to show the complete elution patterns of the three cell loadings.

Figure 5c. This plot may not be readable for color-blind readers. Moreover, the choice of the color green for the diseased condition and red for the control could lead to mis-interpretation of the figure.

We thank the Reviewer for the suggestion. In the revised manuscript, we changed the symbol shapes as well as the fill color in Figure 5c to make it easier to differentiate.

In Figure S6 the text is completely unreadable.

Figure S6 text (Fig. S7 in the revised manuscript) was enlarged in the revised manuscript to make it readable.

Reviewers' comments:

Reviewer #2 (Remarks to the Author):

1. I would like to recollect that 8M urea allows only partial solubilization of membrane proteins. This has been often demonstrated. For example by Masuda et al. J. Proteome Res. 2008, 7, 731-740. Thus a correlation RapiGest vs. 8M urea is worthless.

2. As I have already said: I do not believe that analyses with 1300 proteins identified per sample can provide any valuable insights in human or any eukaryotic proteome.

Reviewer #5 (Remarks to the Author):

In my last review found the reported plots and numbers of the pancreatic islet data confusing. I still have some remarks about the way this is addressed now.

First, unfortunately you only make the data available for the reviewers. I think it would be advantageous to also make the data available as a supplementary table for the reader as well. If this data is available in the PRIDE submission, I'm afraid I could not check that since I could not find any details on how to log in and obtain that data. As a side note, please mention the base of the logarithm you use to transform your p-values.

The combined threshold of both the FDR and the p-value makes it harder to estimate the reliability of the results. The FDR threshold of 0.05 is not necessary: there are still 169 significantly regulated proteins at an FDR threshold of 0.01. This also removes the necessity for the additional p-value threshold.

line 570: "To address potential limitations of data imputation, we also applied an independent G-test with multiple testing correction using the Benjamini-Hochberg (sic) method for those proteins with missing data points."

First, Benjamini-Hochberg is misspelled throughout the article.

Aside from that, I'm not complete sure how to interpret the results of these analyses, since a test is performed on a subset of the data that was already deemed significant in a different test, after which half the numbers were left. I recommend to show the impact of the imputation by using the same type of test on the data with or without the imputed values, which could be the G-test in this example, in a table that is included as supplementary data. It would also be interesting to see

how the direction of regulation correlates with the reported ratio, since these two values correlate but do not correspond 100% (see included chart).

One more thing is that the caption of figure S13 still reads "coefficients of variance", maybe this should be checked again.

Point-by-Point Response to the Reviewer Comments

Reviewer comments in italics, responses in blue.

Reviewer #2 (Remarks to the Author):

1. I would like to recollect that 8M urea allows only partial solubilization of membrane proteins. This has been often demonstrated. For example by Masuda et al. J. Proteome Res. 2008, 7, 731-740. Thus a correlation RapiGest vs. 8M urea is worthless.

This comment has been addressed previously and refers to a very minor point of the manuscript (Supplementary Figure 15) in which we demonstrate that the extraction efficiency of the nanoPOTS platform is similar to widely used protocols including 8M urea. We acknowledge that membrane protein studies are always a challenge for the proteomics field. Our conclusion is only that our approach performs similarly towards membrane proteins compared to traditional urea based approaches.

2. As I have already said: I do not believe that analyses with 1300 proteins identified per sample can provide any valuable insights in human or any eukaryotic proteome.

We disagree with the reviewer on this point. First, we did identify and quantify far more than 1300 protein groups even for our smallest samples. Second, we demonstrate that statistically significant differences in protein expression for hundreds of protein groups can be determined for nanoscale clinical samples (single pancreatic islet slices in control vs. type 1 diabetes donors). The biological significance of the differentially expressed proteins is also discussed; Third, the coupling of laser microdissection with nanoPOTS has the potential to perform spatially resolved proteome mapping, which will provide much more chemical information in measurements than traditional imaging approaches.

Reviewer #5 (Remarks to the Author):

In my last review found the reported plots and numbers of the pancreatic islet data confusing. I still have some remarks about the way this is addressed now.

First, unfortunately you only make the data available for the reviewers. I think it would be advantageous to also make the data available as a supplementary table for the reader as well. If this data is available in the PRIDE submission, I'm afraid I could not check that since I could not find any details on how to log in and obtain that data. As a side note, please mention the base of the logarithm you use to transform your p-values.

We agree with the reviewer and thank him for the suggestion. We now include the data as “Supplementary Data 1”. The legend of the Data was provided (see cover letter). To avoid the confusion of p-value, we have reported the original p-value in the Supplemental Data 1 without log transformation.

The combined threshold of both the FDR and the p-value makes it harder to estimate the reliability of the results. The FDR threshold of 0.05 is not necessary: there are still 169 significantly regulated proteins at an FDR threshold of 0.01. This also removes the necessity for the additional p-value threshold.

We thank the reviewer for the suggestion. In the revised analysis, we only applied a q-value cutoff of 0.02 (based on the Benjamini-Hochberg procedure) and fold-change >2 to identify significant proteins. The additional p-value threshold was removed.

line 570: "To address potential limitations of data imputation, we also applied an independent G-test with multiple testing correction using the Benjamini-Hochberg (sic) method for those proteins with missing data points."

First, Benjamini-Hochberg is misspelled throughout the article.

Aside from that, I'm not complete sure how to interpret the results of these analyses, since a test is performed on a subset of the data that was already deemed significant in a different test, after which half the numbers were left. I recommend to show the impact of the imputation by using the same type of test on the data with or without the imputed values, which could be the G-test in this example, in a table that is included as supplementary data. It would also be interesting to see how the direction of regulation correlates with the reported ratio, since these two values correlate but do not correspond 100% (see included chart).

We apologize the confusion. We acknowledge that it is not trivial to evaluate the impact of missing-data imputation. We have followed the reviewer's suggestion in the revised analysis in which a single test (student's t-test) with the Benjamini-Hochberg procedure for FDR control was applied with and without missing data imputation (See Supplemental Data 1). Just as predicted by the Reviewer, the missing data imputation does help to identify more significant proteins. Also as predicted by the reviewer, the ratio values do not exactly correspond 100%, but they are correlated with $R^2 \sim 0.95$ (Supplementary Figure 16b), suggesting that the imputation procedure is largely solid.

One more thing is that the caption of figure S13 still reads "coefficients of variance", maybe this should be checked again.

This has been corrected. We again sincerely thank the reviewer for the valuable suggestions.